# `Mozart`: Modularized and Efficient MoE Training on 3.5D Wafer-Scale Chiplet Architectures

Shuqing Luo[†1], Han Ye[†2], Pingzhi Li[†1], Jiayin Qin[†2]
Jie Peng[1], Yang (Katie) Zhao[2], Yu (Kevin) Cao[2], Tianlong Chen[*1]

[1]University of North Carolina at Chapel Hill   [2]University of Minnesota - Twin Cities

## Abstract

Mixture-of-Experts (MoE) architecture offers enhanced efficiency for Large Language Models (LLMs) with modularized computation, yet its inherent sparsity poses significant hardware deployment challenges, including memory locality issues, communication overhead, and inefficient computing resource utilization. Inspired by the modular organization of the human brain, we propose `Mozart`, a novel algorithm-hardware co-design framework tailored for efficient training of MoE-based LLMs on 3.5D wafer-scale chiplet architectures. On the algorithm side, `Mozart` exploits the inherent modularity of chiplets and introduces: (1) an expert allocation strategy that enables efficient on-package all-to-all communication, and (2) a fine-grained scheduling mechanism that improves communication-computation overlap through streaming tokens and experts. On the architecture side, `Mozart` adaptively co-locates heterogeneous modules on specialized chiplets with a 2.5D NoP-Tree topology and hierarchical memory structure. Evaluation across three popular MoE models demonstrates significant efficiency gains, enabling more effective parallelization and resource utilization for large-scale modularized MoE-LLMs.

## 1   Introduction

The human brain, known for its cognitive efficiency and modular organization, has long inspired the design of large-scale computational systems [9, 14, 27]. It comprises specialized modules that handle distinct tasks, ranging from memory-intensive to computation-heavy operations, while maintaining low-latency coordination with adjacent regions [1, 3, 13]. This modularity enables efficient, scalable, and flexible processing [2, 17], which is a principle increasingly adopted in deep learning systems such as Large Language Models (LLMs) [21, 35].

Meanwhile, recent advances in LLMs, particularly Mixture-of-Experts (MoEs), reflect similar modular principles by dynamically activating specialized sub-networks based on input. However, the scale and heterogeneity of MoE-LLMs pose significant challenges for conventional hardware platforms [21] (*e.g.*, traditional GPUs or CPUs), including photoreticle-limited scalability [18] and transistor scaling limits [36], as well as poor memory locality [12], high inter-module communication overhead [15], and inefficient resource utilization [25] due to dynamic and uneven computational workloads.

2.5D/3.5D heterogeneous chiplet-based architectures have gained popularity due to their scalability and modularity to meet the demands of the aforementioned LLM-related workloads, including MoEs [32, 46, 8, 20]. Typically in 2.5D designs, multiple chiplets are interconnected via a Network-on-Package (NoP) through an interposer [7, 34, 26, 37], reducing the area and cost overhead of monolithic integration. To further boost inter-chiplet bandwidth, 3D integration techniques such

---

[*]Correspondence to: Tianlong Chen <tianlong@cs.unc.edu>.
[†]Equal Contribution

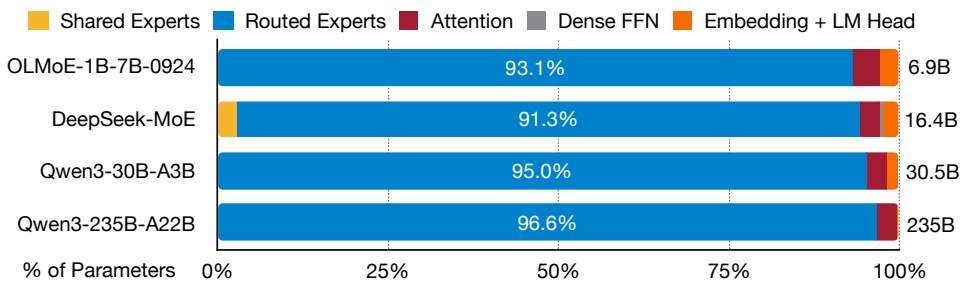

Figure 1: **Parameter distribution in modern MoE-LLMs across various scales**. The routed experts module constitutes over $90\%$ of the total parameters in these architectures.

as Through-Silicon Vias (TSVs) are employed along the vertical direction. However, prior works tend to neglect wafer-scale integration [8, 20, 46] and largely adopt coarse-grained, static workload partitioning strategies [32] that assume dense and uniform computation, tiling the model workload to different chiplets without incorporating system-level coordination and optimization [23]. None of these works consider or explicitly address the system-level design challenges posed by MoE-LLMs with fine-grained modularity, which often results in excessive inter-chiplet communication and inefficient resource utilization.

Given the growing need for system-level coordination in modular LLM deployment, we propose Mozart, an algorithm–hardware co-design framework for efficiently mapping MoE-LLMs onto 3.5D wafer-scale chiplet platforms. Our contributions are summarized as follows:

- To efficiently train MoE-based LLMs, we propose Mozart, a comprehensive framework with optimization techniques including: (1) a *specialized expert layout strategy* placing frequently co-activated experts on the same or adjacent chiplets, targeted at balanced workload across multiple chiplets, (2) a *communication-efficient all-to-all strategy* utilizing expert collaboration, and (3) a *fine-grained scheduling strategy* for improved communication-computation overlap using streaming tokens and experts.

- To better accommodate the modularized structure of MoE, we design a wafer-scale 3.5D chiplet architecture, featuring tightly integrated 3D logic-on-memory stacks, a 2D NoP-tree interconnect, and two-level memory hierarchy. It supports our proposed optimizations with low-latency on-chip activation reuse, communication-aware expert clustering, and inter-leaved execution of communication and computation tailored for sparse MoE computing.

- Mozart achieves over 1.9× acceleration compared to baseline methods when evaluated across three popular open-source MoE-LLMs with varying scales, demonstrating its potential to enhance parallelization efficiency and optimize resource utilization for the post-training deployment of large-scale modularized MoE-LLMs.

## 2 Related Works

**Modularized LLMs.** Modularized LLMs, also known as Mixture-of-Experts (MoE) [35], demonstrate exceptional efficiency on scaling model capacity, allowing significant parameter growth without proportional computational costs. This efficiency derives from replacing traditional dense feed-forward layers with sparse modularized components, where sophisticated routing mechanisms selectively direct input tokens to appropriate expert subnetworks. Models like Mixtral-8x7B [16] demonstrate how activating just two experts per token per layer can leverage a substantially larger parameter space, matching the performance of dense counterparts while dramatically reducing active parameter requirements. The architecture was further refined in DeepSeek-MoE [4, 10], which introduced finegrained experts and shared experts to improve specialization and parameter efficiency. The expert *specialization* phenomenon—where routing networks learn to direct specific input patterns to dedicated experts—enhances processing proficiency [4, 22, 42]. Complementing this, expert *collaboration*, the strategic co-activation of multiple experts for processing complex inputs, has recently minimized communication overhead through optimized expert placement and routing algorithms [24, 47]. In our work, we leverage these expert specialization and collaboration principles to enhance training efficiency specifically for 2.5D/3.5D wafer-scale chiplet architectures, where physical hardware modularity naturally complements the logical modularity of MoE systems.

**2.5D/3.5D Chiplet for ML Workloads.** Chiplet-based architectures have emerged as a promising solution to support the growing computational demands of large-scale neural networks and LLMs.

Prior works such as Maestro [20], Cambricon-LLM [46], and ScalePoM [8] primarily focus on sub-wafer-scale chiplet designs. Maestro adopts a 3D memory-on-logic structure to coordinate multiple small-scale systolic arrays for inference acceleration. Cambricon-LLM integrates Neural Processing Units (NPUs) with flash-based chiplets for energy-efficient on-device inference, while ScalePoM explores hierarchical power delivery for the chiplets. However, these works mainly focus on LLM inference and do not consider wafer-scale integration, limiting their scalability.

In contrast, FRED [32] explores wafer-scale integration by leveraging high-bandwidth interconnects and in-network collective communication to accelerate LLM training. Nonetheless, it largely relies on coarse-grained, static workload partitioning strategies that assume dense and uniform computation. When applied to sparse and modular models such as MoEs, such strategies result in inefficient resource utilization and increased inter-chiplet communication. To overcome these limitations, we propose a 3.5D heterogeneous chiplet architecture that combines vertical stacking with 2.5D NoP-Tree interconnects, providing high-bandwidth, energy-efficient communication while maintaining architectural modularity. Built upon this hardware foundation, we introduce a fine-grained modular partitioning and communication-aware scheduling framework tailored for the post-training process of sparse workloads like MoE. By aligning expert activation patterns with the chiplet topology, our design reduces redundant data movement and significantly improves system throughput under modular model execution.

## 3 Preliminary

### 3.1 Mixture-of-Experts

**Formulation.** Given an input token embedding $\boldsymbol{x}$, the output of an MoE layer can be formulated as the weighted sum of outputs from the $N_e$ experts $\{E_0, E_1, \ldots, E_{N_e-1}\}$:

$$\texttt{MoE}(\boldsymbol{x}) = \sum_{i=0}^{N_e-1} \mathcal{R}(\boldsymbol{x})_i \cdot E_i(\boldsymbol{x}), \tag{1}$$

where $\mathcal{R}(\boldsymbol{x})_i$ is the output of a small gating network $\mathcal{R}(\cdot)$ for the $i$-th expert. For each token, the MoE layer aggregates the output of $k$ experts, determined by the indices of the top-$k$ highest routing scores, derived from the *Softmax* value of a gating function $g(\cdot)$, which is usually a single linear layer:

$$\mathcal{R}(\boldsymbol{x}) = \texttt{top-k}(Softmax(g(\boldsymbol{x})), k) \tag{2}$$

**Expert Parallelism Pipeline.** Expert parallelism [11, 21] has been demonstrated to be the most efficient distributed training technique for MoE models, where different experts are scattered on different parallel units and the workloads are dispatched to each unit during both forward and backward pass. Specifically, a typical MoE pipeline with expert parallelism in the forward pass can be formulated as *Dispatch -> All-to-All -> Expert Computing -> All-to-All -> Combine* [15].

### 3.2 Analyzing Expert Activation Prior

`Mozart` focuses on efficient post-training of MoE-LLMs on chiplet systems. Before deployment, we first analyze the empirical prior of the routing policy and then develop scheduling algorithms to enhance post-training efficiency. Given an instruction tuning dataset, we first run the prefilling stage of inference on it to get the routing choice of a large token batch $\mathcal{B}$, and next we compute 2 metrics:

**Analyzing Workload Distribution across Individual Experts.** We construct a vector $\mathcal{V}$ with $N_e$ elements to quantify the workload distribution across individual experts, where

$$\mathcal{V}_i = \sum_{\boldsymbol{x} \in \mathcal{B}} \mathbb{1}\{\mathcal{R}(\boldsymbol{x})_i \neq 0\}, \quad \mathcal{V}_i = \mathcal{V}_i / \sum_{j=0}^{N_e} \mathcal{V}_j. \tag{3}$$

**Analyzing Collaboration Pattern across Paired Experts.** To uncover co-activation patterns among experts in a single MoE layer, we construct a graph $\mathcal{G}$ with $N_e$ nodes. This graph is represented by an adjacency matrix $\mathcal{C} \in \mathbb{R}^{N_e \times N_e}$, where $\mathcal{C}_{i,j}$ denotes the edge value between nodes $i$ and $j$. We further normalize it with the maximum edge value to confine all entries in the matrix to the interval $[0, 1]$:

$$\mathcal{C}_{i,j} = \sum_{\boldsymbol{x} \in \mathcal{B}} \mathbb{1}\{\mathcal{R}(\boldsymbol{x})_i \neq 0 \wedge \mathcal{R}(\boldsymbol{x})_j \neq 0\}, \quad \mathcal{P}_{i,j} = \mathcal{C}_{i,j} / \max_{0 \leq i,j \leq N_e-1} \mathcal{C}_{i,j}. \tag{4}$$

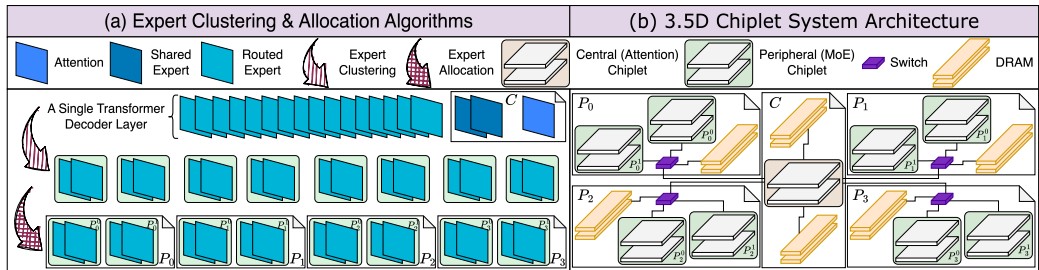

Figure 2: **Algorithm-Hardware Co-Design Diagram of** Mozart. Mozart provides an algorithm-hardware co-design approach, and we present both the *algorithm-level* expert clustering & allocation schemes in the left part, and the *architecture-level* 3.5D chiplet system in the right part. The MoE-LLM parameters are modularized in each decoder layer and mapped to the individual chiplets.

## 3.3 Efficient All-to-All Communication

*All-to-All* communication is a key bottleneck in expert parallelism, as it necessitates synchronization across all parallel units—a process often constrained by communication bandwidth. Reducing the data volume in such communication can effectively reduce end-to-end latency. In this paper, we quantify the communication complexity using the average number of replications per token in the *Dispatch* stage, denoted as $\mathcal{C}_\mathcal{T}$. We prove in the Appendix that $\mathcal{C}_\mathcal{T}$ is the least upper bound for the ratio between the actual data volume during all-to-all communication and the number of tokens in a training step. In standard expert parallelism frameworks [12], $\mathcal{C}_\mathcal{T} = k$ under top-$k$ routing. However, if two co-activated experts for a token are assigned to the same parallel unit (*e.g.*, a GPU in modern data centers or a chiplet in Mozart), only one replica would be required, therefore reducing $\mathcal{C}_\mathcal{T}$. By optimizing expert layout to increase the likelihood of such co-location, $\mathcal{C}_\mathcal{T}$ can be further minimized, thereby lowering the overhead of all-to-all communication.

## 4 Methodology

### 4.1 Overview of Mozart

We detail the design principles and methodology of Mozart with an overview in Figure 2, which addresses key bottlenecks of the post-training process of MoE-LLMs on chiplet systems through algorithm–hardware co-design.

On the algorithm side, we first profile the instruction tuning dataset using the pre-trained model, then apply strategic optimizations to improve post-training efficiency: ❶ **Expert Clustering and Allocation**: We cluster individual experts using the collaboration pattern prior, and map these clusters to chiplets using the workload distribution prior, aiming at balancing workload across MoE chiplet groups. More details are provided in Sec. 4.2. ❷ **Fine-grained Scheduling**: To overlap the DRAM communication overhead with on-chip computing, we propose streaming both the expert loading process and expert computing process of tokens using fine-grained scheduling, following the expert layout derived from the clustering and allocation algorithms. More details are provided in Sec. 4.3.

From the hardware side, we propose a 3.5D wafer-scale chiplet architecture featuring: ❶ **2.5D NoP-Tree Topology**: We propose the 2.5D NoP-tree interconnect in Mozart that organizes attention chiplets as central dispatchers and expert chiplets as leaves. Switches enable in-network MoE aggregation, reducing communication latency and bandwidth cost. ❷ **Hierarchical Memory Structure**: Mozart introduces a two-level memory structure with model weights stored in distributed DRAM and activations cached in local SRAM. To further reduce data access latency, we adopt a logic-on-memory 3D integration, where each compute chiplet vertically stacks a compute die with an SRAM die via hybrid bonding. This tightly coupled design enables fast local access to intermediate results, such as activations, and aligns well with their temporal reuse patterns. More details are provided in Sec. 4.4.

### 4.2 Expert Collaboration for Efficient On-Package All-to-All Communication

Although every expert may be activated, the activation and co-activation patterns are not exactly balanced in practice. We take the profiling results on Alpaca [40] using DeepSeek-MoE [4] as an example. At the final layer, some experts are sensibly activated more frequently (long horizontal bar in Figure 3), and

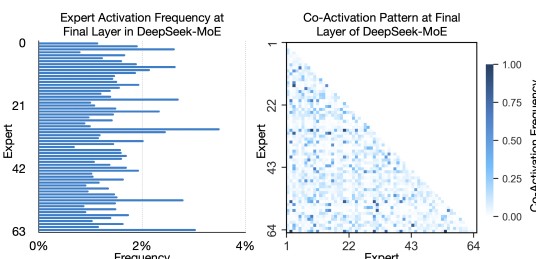

Figure 3: Left: Activation frequency for pre-trained DeepSeek-MoE, indicating *expert specialization*. Right: Co-activation pattern for pre-trained DeepSeek-MoE, indicating *expert collaboration*.

some expert pairs are also activated more frequently (dark-colored blocks in Figure 3), motivating us to specialize the expert layout on chiplets for balanced workload distribution during post-training. This clustered layout can also reduce the all-to-all communication volume, which is synchronous and cannot be overlapped with computation. To determine the expert placement on chiplets, we implement a 2-stage approach as follows.

**Stage-1: Expert Clustering.** We cluster individual experts as candidates for expert-chiplet assignment, aiming at enhancing intra-cluster collaboration while minimizing inter-cluster collaboration. Intra-cluster collaboration is defined as the average co-activation frequency among all expert pairs in a single cluster, whereas inter-cluster collaboration represents the average co-activation frequency between all expert pairs across 2 distinct clusters. Inspired by the farthest point sampling algorithm in point cloud learning [31], we implement the clustering as shown in Algorithm 1.

**Stage-2: Expert Cluster Allocation.** Since our 3.5D chiplet architecture (Figure 2) allocates a DRAM chip for a *group* of MoE chiplets interconnected with a switch, balanced workload distribution across these *group*s becomes critical. To achieve this, we formalize the cluster-chiplet assignment as a binary integer programming problem. Let $N_g$ denotes the number of *group*s (asserting $N_c$ can be divided by $N_g$) and a binary matrix $\mathcal{M} \in \{0,1\}^{N_g \times N_c}$ represents the cluster-*group* assignment, our optimization objective is formulated as:

$$\min_{\mathcal{M}} |\mathcal{MV} - \mathcal{V}_{aux}|, \ \text{s.t.} \sum_{i=0}^{N_g} \mathcal{M}_{[i,j]} = 1, \ \forall \, 0 \le j \le N_c \ \text{and} \ \sum_{j=0}^{N_c} \mathcal{M}_{[i,j]} = 1, \ \forall \, 0 \le i \le N_g, \quad (5)$$

where $\mathcal{V}_{aux}$ is an auxiliary vector with $N_g$ elements, each one equals to $1/N_g$.

---

**Algorithm 1 Expert Clustering.**

---

**Require:** Adjacent matrix $\mathcal{C} \in \mathbb{R}^{N_e \times N_e}$ for graph $\mathcal{G}$, number of chiplets $N_c$ (also the number of clusters).
  **Initialize** expert clustering result $\mathcal{L}$ with $N_c$ empty lists.
  **for** $c \leftarrow 0, N_c - 1$ **do**
    **if** $c == 0$ **then**
      Find the 2 most highly co-activated experts, and push them into $\mathcal{L}_{[c]}$.
    **else**
      Find an unselected expert with the lowest co-activation frequency with the experts in $\mathcal{L}$.
      Push it into $\mathcal{L}_{[c]}$.
    **end if**
    **while** $\text{len}(\mathcal{L}_{[c]}) \le N_e/N_c$ **do**     ▷ Assert $N_e$ can be divided by $N_c$.
      Find an unselected expert with the highest average co-activation frequency with the experts in $\mathcal{L}_{[c]}$.
      Push it into $\mathcal{L}_{[c]}$.
    **end while**
  **end for**
  **return** $\mathcal{L}$.

---

### 4.3 Fine-grained Scheduling with Streaming Tokens and Experts

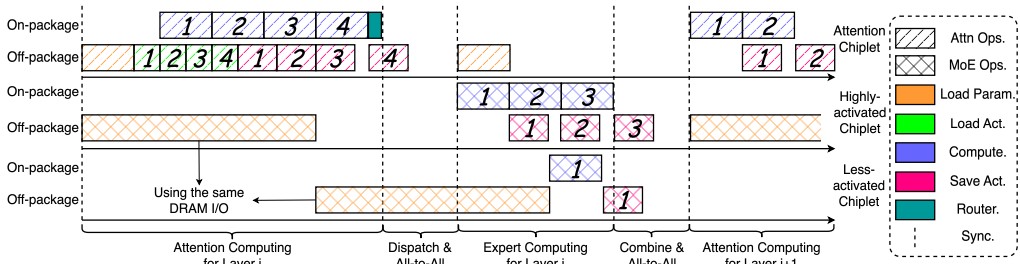

Figure 4: **Fine-Grained scheduling pipeline in the forward pass**. The streaming tokens, marked with the execution order, can effectively overlap the computation (purple blocks) and DRAM communication (pink blocks, saving activations). We present 3 types of chiplets in the training pipeline, including attention chiplet, highly-activated chiplet, and less-activated chiplet. Since the 2 MoE chiplets share the same DRAM I/O, the highly activated experts should be first loaded to the chiplet for better communication-computation overlap.

The sheer size of parameters in MoE-LLMs necessitates storing them in DRAM and dynamically loading layers to chiplets for computation. However, this approach incurs significant communication overhead compared to on-chip processing. To mitigate this bottleneck and enhance training parallelism, we propose a *fine-grained scheduling* scheme through streaming experts and tokens:

**Streaming Experts** Since multiple MoE chiplets within a *group* share the same DRAM, their concurrent memory accesses require serialization. To optimize parallelism, we strategically schedule communication order by ranking expert clusters: using profiled workload distribution $\mathcal{V}$, we quantify the importance of an expert cluster using the aggregated per-expert workloads, and prioritize the loading order of expert clusters with heavier computational workload first.

**Streaming Tokens** Partitioning the global token batch into streaming tokens (micro-batches) enables overlapping DRAM communication (for saving activations during backward passes) with on-chip computation. To be specific: (1) For the attention module, all tokens are partitioned into *streaming attention tokens*; (2) For the MoE module, the workload of each expert is partitioned into *streaming expert tokens*, and different experts on the same chiplet are computed sequentially.

**Fine-Grained Scheduling** The huge routed experts (Figure 1) results in significant communication overhead between DRAM and MoE chiplets, so we overlap it with on-chip computations using fine-grained scheduling in Figure 4, which mainly occurs in 2 aspects: (1) Loading highly-activated cluster & Attention computing; (2) Loading less-activated cluster & Highly-activated cluster computing.

### 4.4 Wafer-Scale Chiplet Architecture

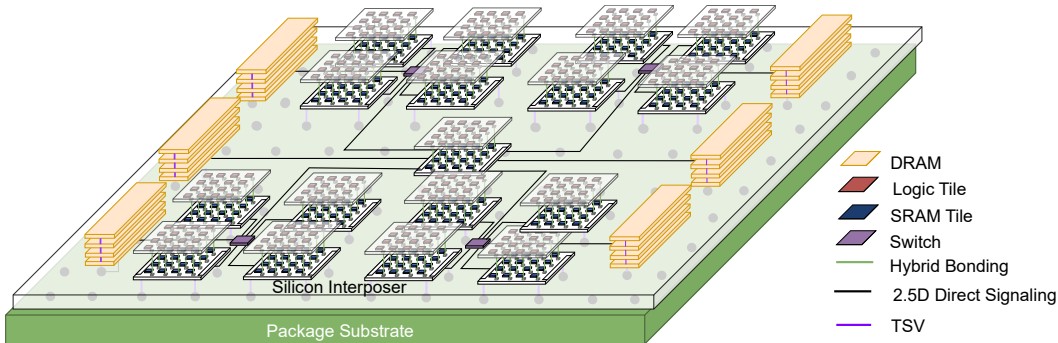

Figure 5: **The overall 3.5D chiplet architecture in** `Mozart`. The hardware architecture implements a three-layer hierarchical tree topology, comprising a central attention node, switch nodes, and peripheral MoE nodes. The two-tier dies are connected face-to-face.

**Motivation of `Mozart` Architecture** Each MoE-LLM decoder block generally involves 2 stages: (1) the attention module and router network, and (2) expert computation with structural sparsity. This heterogeneous and dynamic execution flow poses significant challenges to system-level scheduling and communication. To tackle them, we propose a wafer-scale 3.5D chiplet architecture that integrates heterogeneous compute and memory resources: (1) To improve **memory locality**, we design a hierarchical memory system aligned with the temporal reuse patterns of MoE-LLMs, enabling frequently reused data such as activations to be cached closer to the computing unit using the 3D logic-on-memory stack, thereby reducing costly accesses to off-chip DRAM; (2) To reduce **communication overhead across chiplets**, we co-locate frequently co-activated experts onto the same chiplet based on profiling of activation patterns, significantly reducing costly inter-chiplet transfers during both forward and backward passes; (3) To enhance **compute resource utilization**, we disaggregate memory-bound and compute-bound components onto specialized chiplets with matching bandwidth and compute capabilities, and apply fine-grained, pipeline-aware scheduling to balance load and overlap communication with computation.

**Physical Layout** To efficiently support the computation flow, we introduce a wafer-scale chiplet architecture that combines 3.5D integration with a 2.5D NoP-tree interconnect, as shown in Figure 5. ❶ **3D Chiplet Stack:** Each computing chiplet integrates a logic die and an SRAM die in a vertical stack using hybrid bonding, supporting either attention or MoE-based FFN operations. The SRAM layer serves as a fast buffer for intermediate data with frequent reads and writes. Leveraging the short vertical interconnects enabled by 3D integration, this tightly coupled logic-on-memory chiplet offers significantly higher bandwidth and lower latency compared to conventional 2D designs, as illustrated in Figure 5. ❷ **2.5D NoP-Tree Topology:** The inter-chiplet network adopts a 2.5D NoP-Tree

topology [19] that disaggregates attention and MoE operations across the network. Memory-bound attention chiplets are placed near the center of the tree as dispatching nodes with higher DRAM bandwidth, while compute-bound expert-cluster chiplets are organized as leaf nodes to execute MoE feed-forward computations with more computing resources. The system comprises 16 MoE chiplets, partitioned into 4 switch-connected groups, each containing 4 expert-cluster chiplets with pre-defined placement. Moreover, the switch modules are equipped with in-network compute capabilities to aggregate MoE outputs locally, significantly reducing inter-chiplet communication and improving pipeline throughput. ❸ **Memory Hierarchy:** We design a two-level memory hierarchy for `Mozart`. Model weights are stored in DRAM distributed around the core on the wafer. Every four expert clusters share a dedicated DRAM I/O interface. The DRAM connects to SRAM on the attention chiplet and switches for the MoE groups, enabling weight transformation from off-chip to the computing unit. Given that weights are relatively static during one iteration of training and exhibit low temporal access locality, they are suited for off-chip DRAM storage. In contrast, activations are highly transient and frequently accessed during the computation pipeline. Therefore, they are cached in the local SRAM die under each computing die to minimize access latency and support rapid data exchange during the process.

**Algorithm-to-Hardware Mapping** To efficiently execute the MoE-LLM models on the proposed chiplet-based architecture, we map its major computational components, including attention and MoE-expert layers, onto specialized chiplets with coordinated dataflows and scheduling strategies. ❶ **Dataflow of the Training Process**: During each training step, the system processes 32 samples (sequences), divided into 4 serially executed micro-batches of size 8. A weight-streaming strategy is adopted, where only one transformer block's weights are loaded at a time, in response to area and interconnect constraints of the wafer-scale architecture. QKV projection and multi-head attention score computation are mapped to multiple systolic arrays (SAs). SAs are grouped into tiles, each integrating a local adder tree to aggregate partial sums and reduce intermediate communication. These partial results are transmitted from the compute die and stored in the underlying SRAM die via hybrid bonding. **After attention completes**, activations are routed through the NoP-Tree network to a switch module with for token-wise routing and reduction. Tokens are dispatched to selected experts for FFN. Local aggregation is performed within each expert-cluster chiplets, followed by global aggregation via the switch. The aggregated expert outputs are routed back to the attention module to continue processing the subsequent transformer blocks. During backpropagation, gradients follow the reverse path, with parameter updates performed locally on attention and expert chiplets before being written back to DRAM. ❷ **Scheduling for Computing-Communication Pipeline**: To improve compute throughput and resource utilization, the system adopts fine-grained pipeline scheduling. Leveraging the temporal locality in expert selection across adjacent training steps, frequently activated weights are prefetched onto expert chiplets ahead of token routing, reducing memory stalls. **At runtime**, each switch group coordinates micro-batch-level pipelined execution. Based on the received activation load per token, chiplets within a group sequentially fetch weights from DRAM via the shared switch. While one micro-batch undergoes FFN computation, the next micro-batch's weights are concurrently loaded, enabling overlapped execution of compute and memory access. A similar strategy is applied during backpropagation to hide communication latency and sustain high training throughput.

## 5 Experiments

### 5.1 Algorithmic Setup

Table 1: **Configurations of three pre-trained MoE-LLMs used in our experiments**.

| Model | # Total Parameters | # Activated Parameters | # Routed Experts | # Shared Experts | Hidden Size | # Layers | Routing |
|---|---|---|---|---|---|---|---|
| Qwen3-30B-A3B [41] | 30.5B | 3.3B | 128 | 0 | 2048 | 48 | top-8 |
| OLMoE-1B-7B-0924 [28] | 6.92B | 1.3B | 64 | 0 | 2048 | 16 | top-8 |
| deepseek-moe-16b-base [4] | 16.4B | 2.7B | 64 | 2 | 2048 | 28 | top-6 |

Our experiments include three MoE models with various architectures: `Qwen3-30B-A3B` [44, 45], `OLMoE-1B-7B-0924` [28], and `deepseek-moe-16b-base` [4]. Details of them are summarized in Table 1. We use Alpaca [40], an instruction tuning dataset of 52K samples, for all our experiments. Our evaluation includes *latency* and *energy* as metrics to indicate the real-world impact of our designs. We use NVIDIA A100 80G GPU servers and PyTorch for our profiling and simulation experiments.

Table 2: **Hardware metrics of the three MoE-LLMs used in our experiments**. The number of inter-chiplet links is computed based on the chiplet area for 2.5D signaling. The link counts are calculated as the product of horizontal and vertical link numbers for the 3D stack. The total area encompasses not only chiplets but also off-chip components such as DRAM.

| Model | Total | | Memory (DRAM/Stack & SRAM/Tile) | | 2.5D Direct Signaling /Link | | 3D Hybrid Bonding /Link | |
|---|---|---|---|---|---|---|---|---|
| | Area (mm$^2$) | Power (kW) | Cap. (MB) | BW (GB/s) | BW (GB/s) | Pitch ($\mu$m) | BW (GB/s) | Pitch ($\mu$m) |
| Qwen3-30B-A3B | 14175 | 3.34 | 8192&2.265 | 256&32 | 0.125 | 50 | 0.125 | 50 |
| OLMoE-1B-7B-0924 | 10200 | 3.55 | 8192&2.265 | 256&32 | 0.125 | 50 | 0.125 | 50 |
| deepseek-moe-16b-base | 11230 | 3.19 | 8192&2.265 | 256&32 | 0.125 | 50 | 0.125 | 50 |

## 5.2 Hardware Setup

The overall `Mozart` architecture comprises 16 expert-cluster chiplets for MoE computation, organized into 4 switch-connected clusters, as well as one dedicated attention chiplet. Each MoE/attention chiplet has 36–100 tiles, with 16 Systolic Arrays (SAs) in one tile and 256–576 Processing Elements (PEs) in one SA. Off-chip memory is provided by 6 HBM2-based DRAM [29], with 4 shared across expert-cluster groups (one per group) and 2 exclusively connected to the attention chiplet to provide high-bandwidth. We implement the logic dies, SRAM dies, inter-chiplet interconnects and switches in Verilog, and synthesize the gate-level netlist using Synopsys Design Compiler [38] targeting 28nm technology. The typical power consumption is as reported by Synopsys PrimePower [39] based on the generated gate-level netlist. To evaluate the performance of `Mozart`, we further develop a cycle-accurate simulator, whose runtime and power outputs are validated against the Verilog simulation results to ensure accuracy. For real-world implementation, we adjust hardware configurations for all three algorithmic baselines with FP16 precision to meet key 3.5D chiplet process constraints. We simulate all the design under 1GHz clock frequency. Detailed configurations for the three models are summarized in Table 2.

## 5.3 Experimental Results

**Effectiveness of the Optimization Techniques**    Table 3 summarizes four configurations of `Mozart` used to evaluate the effectiveness of our proposed algorithm-side methodologies, including one baseline without any optimizations and three variants that incrementally incorporate the optimization methods described in Sections 4.2 and 4.3. The simulation results demonstrate that our proposed 3 optimization techniques can jointly reduce the end-to-end post-training latency, with a $1.92\times$ speedup for `Qwen3-30B-A3B-Base`, $2.37\times$ for `OLMoE-1B-7B-0924` and $2.17\times$ for `DeepSeek-MoE-16B-Base`. We further provide Table 4 to demonstrate the correlation between all-to-all communication data volume and the end-to-end training latency, where `Mozart-A`, B, and C present different data volumes during all-to-all communication, which is positively correlated to latency.

**Study on the Impact of Sequence Length and DRAM bandwidth**    As shown in Figure 6(b), the training latency increases as the sequence length per batch grows from 128 to 512. Although the number of batches decreases accordingly, each micro-batch carries longer sequences and heavier computation loads, which, when executed sequentially, become more constrained by communication bandwidth. This trend is particularly pronounced in the baseline design without any optimizations, where latency rises from 3.88s at length 128 to 7.64s at 512. In contrast, `Mozart-C` consistently achieves the lowest latency across all sequence lengths and exhibits reduced sensitivity to longer sequences, achieving a speedup of $2.34\times$ at sequence length 512 and $1.47\times$ at length 128 compared to the baseline. This improvement stems from its architecture that enables efficient communication-computation overlap and alleviates communication congestion through expert-aware layout and routing, which together mitigate the latency increase caused by longer and heavier micro-batches.

Table 3: **Configurations for different settings used in our experiments**.

| Optimization Technique                     Method | Baseline | Mozart-A | Mozart-B | Mozart-C |
|---|---|---|---|---|
| Specialized Expert Layout on Chiplets (Section 4.2) | ✗ | ✗ | ✗ | ✓ |
| Efficient All-to-All Communication (Section 4.2) | ✗ | ✗ | ✓ | ✓ |
| Communication-Computation Overlap (Section 4.3) | ✗ | ✓ | ✓ | ✓ |

When it comes to study of DRAM bandwidth depicted in Figure 6(c), all configurations achieve lower latency with HBM2 (256GB/s) [29] compared to SSD (15.8GB/s) [43] due to its higher memory

bandwidth. Notably, the relative speedup from Mozart optimizations becomes higher with HBM2 than SSD. This can be attributed to the domination of latency caused by DRAM-based expert weight streaming when using SSD, which remains the bottleneck even after optimization. Since MoE computation accounts for only a small portion of the overall training time, pipelining and token-level scheduling have limited impact when memory access is slow. Furthermore, the communication cost reduced by all-to-all optimization is only about one-third of the streaming latency, making the total gain under SSD more constrained. In contrast, with HBM2, faster streaming allows better utilization of compute-communication overlap, enabling the co-design techniques in Mozart to take full effect.

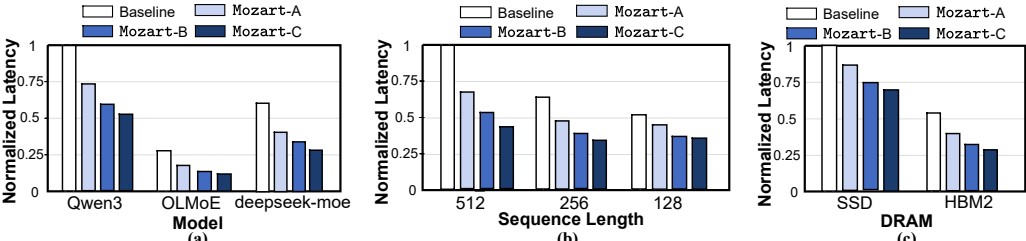

Figure 6: **Experimental results on the chiplet system of** Mozart. We report the average training latency per step for $1k$ iterations, with the micro batch size for *streaming attention/expert tokens* set to $8$. (a) Study on the proposed optimization results (Sequence Length=256, DRAM=HBM2), with a max latency of $\underline{4.87}$ s. (b) Study on the impact of the sequence length on Qwen3-30B-A3B Model (DRAM=HBM2), with a max latency of $\underline{7.65}$ s. (c) Study on the impact of the DRAM bandwidth on Qwen3-30B-A3B Model (Sequence Length=256), with a max latency of $\underline{9.17}$ s.

Table 4: **The corelation between all-to-all communication complexity $\mathcal{C}_{\mathcal{T}}$ and end-to-end latency**. $\mathcal{C}_{\mathcal{T}}$ is calculated by averaging both the training iterations and the MoE layers for each setting.

| Method
Metric | Qwen3-30B-A3B-Base | | | OLMoE-1B-7B-0924 | | | DeepSeek-MoE-16B-Base | | |
|---|---|---|---|---|---|---|---|---|---|
| | Mozart-A | Mozart-B | Mozart-C | Mozart-A | Mozart-B | Mozart-C | Mozart-A | Mozart-B | Mozart-C |
| Normalized Latency | 0.73 | 0.59 | 0.52 | 0.63 | 0.48 | 0.422 | 0.67 | 0.56 | 0.46 |
| $\mathcal{C}_{\mathcal{T}}$ | 8 | 6.58 | 5.77 | 8 | 6.84 | 5.63 | 6 | 5.56 | 4.32 |

## 5.4 Further Investigation

**Q1:** Is Mozart **memory-bound or computing-bound?** **A: Memory-bound**. This is because our proposed 3.5D chiplet architecture in Mozart can well-parallelize the MoE computation workload. While this design successfully eliminates the computational bottleneck associated with heavy MoE operations, the system's overall latency becomes constrained by the sequential MoE weight loading process. This fundamental limitation persists because weight loading throughput cannot be substantially improved without hardware resource upgrades. Consequently, Mozart's performance is primarily governed by this unavoidable sequential bottleneck inherent to current hardware constraints.

**Q2:** **Which algorithmic deigns are more critical in** Mozart**?** **A: Communication-Computation Overlap > Efficient All-to-All Communication > Specialized Expert Layout on Chiplets**. The key insights are: ❶ The communication overhead between DRAM and chiplets is the main bottleneck, and applying it on the baseline can offer $\underline{1.33\times}$ acceleration on Qwen3-30B-A3B-Base, $\underline{1.58\times}$ on OLMoE-1B-7B-0924, and $\underline{1.49\times}$ on DeepSeek-MoE-16B-Base. ❷ The all-to-all communication overhead is a secondary bottleneck for training latency, since it requires synchronization across all the chiplets and is constrained by the on-package bandwidth. Our specialized expert layout on chiplets can further reduce the data volume during all-to-all communication, as we illustrated in Table 4.

**Q3:** Is Mozart **compatible with existing efficient training algorithms?** **A: Yes, it is compatible with parameter-efficient fine-tuning methods such as LoRA, QLoRA,** *etc***.** Mozart' architecture and scheduling mechanisms are designed to work orthogonally to these methods, as they primarily focus on different optimization goals. While PEFT methods reduce the total trainable parameters, Mozart optimizes the physical deployment of MoE workloads on chiplet architectures.

## 6 Conclusion and Limitations

We present Mozart, an algorithm-hardware co-design framework for efficient post-training of MoE-LLMs on chiplet systems. By jointly optimizing expert allocation, fine-grained scheduling, and heterogeneous chiplet mapping on a 3.5D wafer-scale architecture, Mozart significantly improves communication efficiency and hardware utilization, enabling scalable and efficient deployment of modularized workload. While Mozart demonstrates $\underline{1.92\times}$ performance improvement for Qwen3-30B-A3B-Base,

$2.37\times$ for `OLMoE-1B-7B-0924` and $2.17\times$ for `DeepSeek-MoE-16B-Base` on 3.5D wafer-scale architectures, two limitations remain. First, the attention modules are assigned to an individual chiplet, which may lead to suboptimal latency due to limited resources. This can be further tackled with data or tensor parallelism. Second, the switches can become performance bottlenecks under high communication demand. While Mozart currently tries to reduce end-to-end latency through fine-grained scheduling, further improvements may potentially be achieved by allocating more chiplet area to switch resources and increasing bandwidth to achieve low-latency communication.

## Acknowledgement

This research was partially funded by the National Institutes of Health (NIH) under award 1R01EB037101-01. The views and conclusions contained in this document are those of the authors and should not be interpreted as representing the official policies, either expressed or implied, of the NIH. Tianlong Chen was also partially supported by the Amazon Research Award. We thank Nikhil Kumar Cherukuri for the valuable discussions throughout this project.

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

# A  Code

Please find the code base for this paper here: https://github.com/UNITES-Lab/Mozart

# B  More Experimental Results

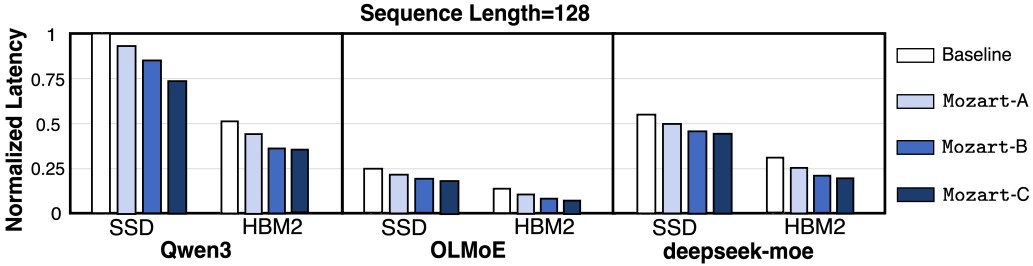

Figure 7: **Normalized Latency Comparison for** 3 **MoE-LLMs with Sequence length** 128. The max wall-clock latency here is 7.61 s (Qwen3 model with baseline method using SSD for DRAM).

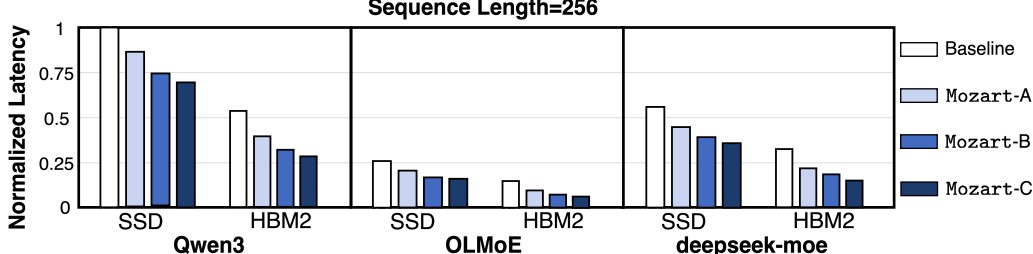

Figure 8: **Normalized Latency Comparison for** 3 **MoE-LLMs with Sequence length** 256. The max wall-clock latency here is 9.17 s (Qwen3 model with baseline method using SSD for DRAM).

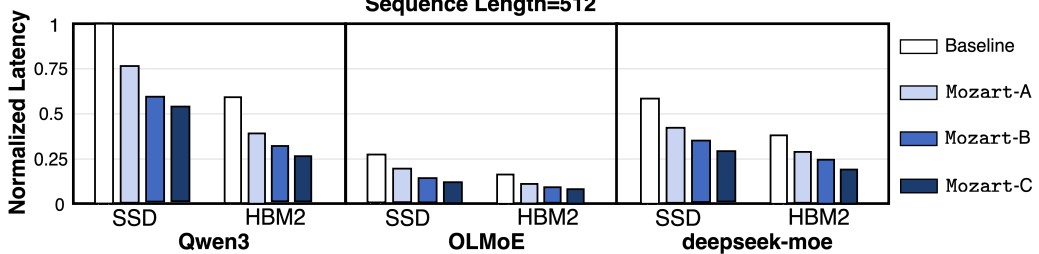

Figure 9: **Normalized Latency Comparison for** 3 **MoE-LLMs with Sequence length** 512. The max wall-clock latency here is 13.03 s (Qwen3 model with baseline method using SSD for DRAM).

We provide comprehensive numerical latency results for all configurations, including 3 sequence length (128, 256, 512), 4 methods (`Mozart` Baseline, `A`, `B`, and `C`), and 2 DRAM (SSD and HBM2). Results comparison visualizations are provided in Figure 7, 8, and 9.

# C  Motivation Explanations

## C.1  Why Attention is Memory-Bound and FFN is Compute-Bound

The chiplet architecture in `Mozart` utilized the fact that in a typical decoder layer in modern large language models, the ***Attention*** module is memory-bound and the ***FFN*** module is computation-bound. We demonstrate it using profiling experiments on the OLMo-2 model series [30]. The experiment settings are:

- We examine a single decoder layer, and collect the wall-clock latency and the FLOPs for both attention and FFN modules.

- The results are collected through running the forward pass, *i.e.*, the prefilling stage of model inference, and the results are normalized for easier comparison.
- We fix the batch size to $4$ and test the sequence length of $512$, $1024$, and $2048$.
- We select OLMo-2 models with $4$ scales, including 1B, 7B, 13B, and 32B.

The profiling experiments are visualized in Figure 10 (1B), Figure 11 (7B), Figure 10 (13B), and Figure 13 (32B). We can find that, the FFN module counts for more FLOPs but less wall-clock latency. It is because the ***Attention*** module is memory-bound and the ***FFN*** module is computation-bound:

- The FFN module counts for more FLOPs because it contains more model parameters. But the computation task for it is mainly composed of large matrix multiplication, which is easy to parallelize. Therefore, the wall-clock latency of it can be lower than attention.
- The Attention module requires frequent memory access operations, which is demonstrated by the Flash-Attention series [6, 5, 33]. Although it contains fewer model parameters, the computation tasks here are difficult to parallelize. Therefore, the attention module counts for more wall-clock latency.

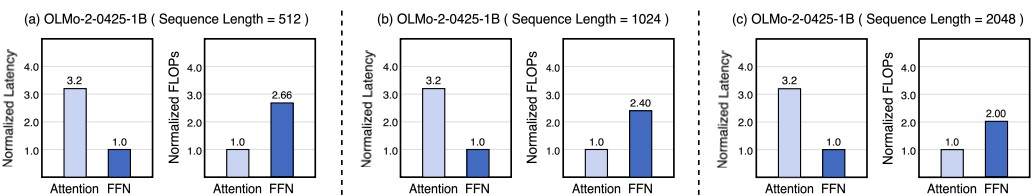

Figure 10: **Profiling results on latency & FLOPs for Attention & FFN using OLMo-2-0425-1B**.

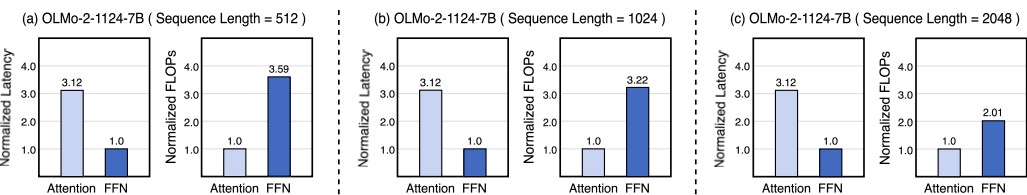

Figure 11: **Profiling results on latency & FLOPs for Attention & FFN using OLMo-2-1124-7B**.

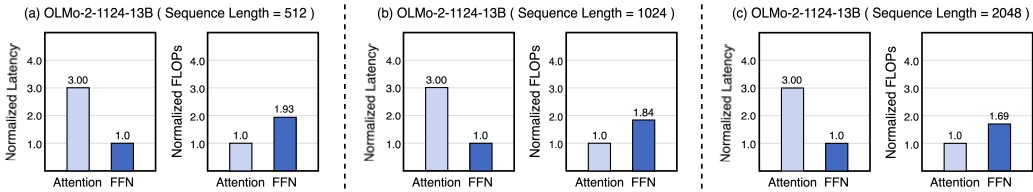

Figure 12: **Profiling results on latency & FLOPs for Attention & FFN using OLMo-2-1124-13B**.

## C.2  Challenges for Mixture-of-Expert Computation

We present 3 challenges for MoE computation in the abstract part of this paper, including memory locality issues, communication overhead, and insufficient computing resource utilization. Our algorithm-hardware co-design scheme in `Mozart` tries to solve these challenges with joint efforts. We demonstrate these challenges through fine-tuning an OLMoE-1B-7B model with $4$-way expert parallelism, with batch size $8$ on each GPU and sequence length $512$. We use MegaBlocks [12], the standard expert parallelism framework, for the MoE modules, and use data parallelism for the attention modules. We employ the dropless MoE implementation. The training speed is 2-3 iterations per second, and we monitor the behavior of each GPU with an interval of $0.1$ s. We take 3 fragments for visualization, as shown in Figure 14, 15, and 16, which demonstrate that both the GPU power and the memory consumption show high dynamism. These phenomena can explain 2 challenges:

- **Memory Locality Issues**: Since the workload for each expert changes dynamically, the activation tensors should be frequently allocated and freed, leading to severe memory management issues.

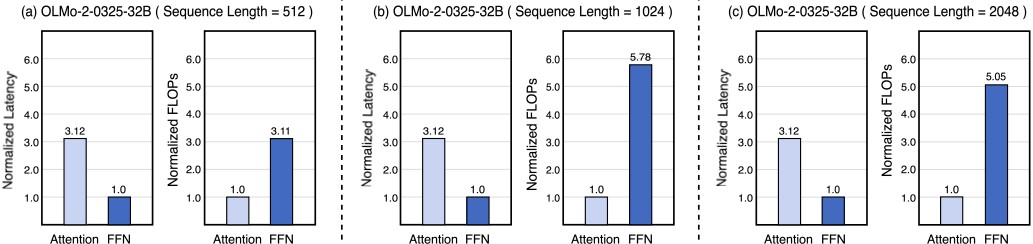

Figure 13: **Profiling results on latency & FLOPs for Attention & FFN using OLMo-2-0325-32B**.

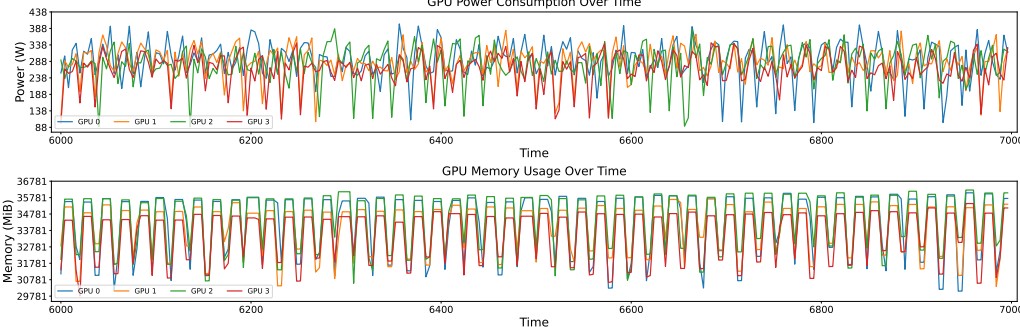

Figure 14: **GPU Behavior Monitor at Time Step 6k-7k**.

- **Insufficient Computing Resource Utilization**: The reason for this challenge lies in 2 aspects: (1) the dynamism of workload leads to dynamism of GPU power, and (2) the dynamism of workload restricts the training batch size to avoid out-of-memory error, which also constrains the utilization of GPU computing resources.

The all-to-all communication issues have been explained in Tutel [15], which is a significant bottleneck for training MoE models at scale, consuming up to $40\%$ of the total runtime.

## D Meassuring All-to-All Communication Complexity with $\mathcal{C}_{\mathcal{T}}$

We propose to measure the all-to-all communication data volume in Section 3.3 using the average replication times of each token, denoted as $\mathcal{C}_{\mathcal{T}}$. We prove that $\mathcal{C}_{\mathcal{T}}$ is the least upper bound of the ratio between actual all-to-all communication data volume and the total number of tokens. We take a single all-to-all communication in $D-$way expert parallelism as an example, and denote the original tokens as $\{\mathcal{S}_i\}_{i=0}^{D-1}$. For a single token $t \in \mathcal{S}_i$ on device $i$, we denote the number of replications for it transmitting from device $i$ to device $j$ as $N_i^j(t)$, *i.e.*, token $t$ on device $i$ activates $N_i^j(t)$ experts preserved on device $j$. In the standard expert parallel framework, given top-$k$ routing, we have

$$\sum_{j=0}^{D-1} N_i^j(t) = k, \ \forall \ t \in \mathcal{S}_i \text{ and } \forall \ 0 \le i \le D-1. \tag{6}$$

For the actual all-to-all communication data volume:

$$
\begin{aligned}
\sum_{i=0}^{D-1} \sum_{t \in \mathcal{S}_i} (\sum_{j=0}^{i-1} N_i^j(t) + \sum_{j=i+1}^{D-1} N_i^j(t)) &\le \sum_{i=0}^{D-1} \sum_{t \in \mathcal{S}_i} (\sum_{j=0}^{i-1} N_i^j(t) + N_i^i(t) + \sum_{j=i+1}^{D-1} N_i^j(t)) \\
&= \sum_{i=0}^{D-1} \sum_{t \in \mathcal{S}_i} (\sum_{j=0}^{D-1} N_i^j(t)) \\
&\le k \cdot \sum_{i=0}^{D-1} |\mathcal{S}_i|
\end{aligned}
\tag{7}
$$

The 2 inequalities in Equation 7 are reached when

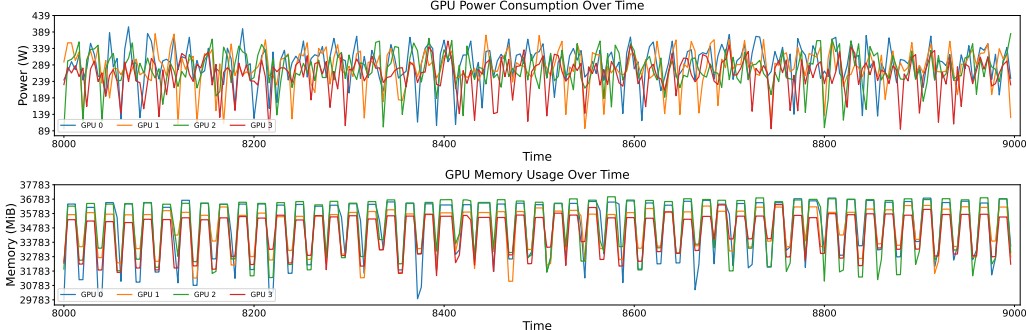

Figure 15: **GPU Behavior Monitor at Time Step 8k-9k**.

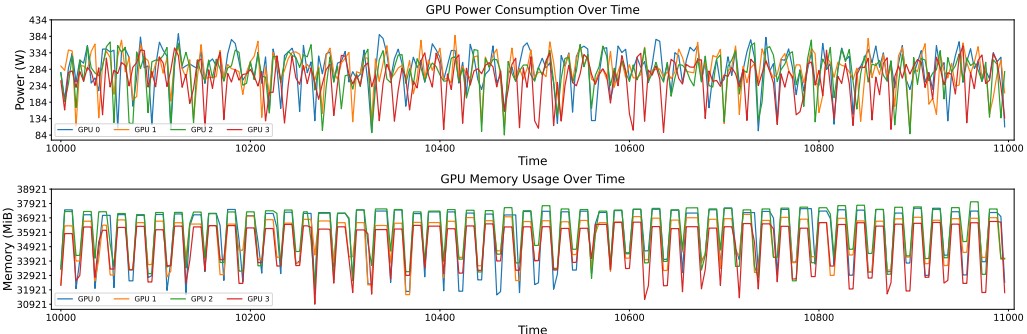

Figure 16: **GPU Behavior Monitor at Time Step 10k-11k**.

- The first one is achieved when $N_i^i(t) = 0$ for all $0 \leq i \leq D-1$ and $t \in \mathcal{S}_i$, *i.e.*, no token would activate the experts kept on the device where the token is originally kept.

- The second one is achieved for standard expert parallelism, *i.e.*, making $k$ replications for each token in the dispatch stage under top-$k$ routing.

The first inequality cannot be utilized for communication efficiency, since it is data-dependent and task-dependent. While the second inequality can be leveraged by employing our proposed strategy in Section 3.3.

## E Impact Statement

As the paper's primary innovation is efficiently deploying the post-training process of MoE-based large language models on the chiplet-based system, it by itself doesn't pose any obvious risks. The potential for negative societal impact depends on the specific MoE-LLMs. We strongly recommend these models be used in compliance with all ethical standards appropriate to the domain in which it is targeted to be deployed.

