# OpenReview forum: "Mozart: Modularized and Efficient MoE Training on 3.5D Wafer-Scale Chiplet Architectures"
_NeurIPS.cc/2025/Conference — NeurIPS 2025 spotlight_

### Official Review · Reviewer_X4mS · 2025-06-09

**Clarity:** 3
**Significance:** 4
**Originality:** 3
**Rating:** 5
**Confidence:** 3

**Summary:**

Mozart is a novel and comprehensive hardware-software co-design framework for training MoE models on 3.5D wafer-scale chiplet architectures. On the software part, Mozart optimizes the expert placement to reduce inter-chiplet communication, and it uses streaming to improve the communication-compute overlap. On the hardware part, Mozart can place heterogeneous components on each chiplet, and it connects them via a 2.5D NoP-Tree.

**Questions:**

Overall, I enjoy reading this paper and have learned a lot!

See weakness 1, the authors can discuss more on why the proposed overlap scheme is better than existing works for this hardware.

To support larger models, how is the proposed solution compatible with existing distributed training optimizations like model parallelism?

**Ethical Concerns:**

["NO or VERY MINOR ethics concerns only"]

**Final Justification:**

All concerns solved in the rebuttal.
This paper well reveal a lot of new knowledge to the community.
However, you may consider directly touching the points in your rebuttal next time, without using too many lengthy descriptions and fancy words.
For the paper itself, I am still positive about it.

**Limitations:**

The paper is great! I can only observe one limitation.

See weakness 2, as LLMs scale quickly, how does a Mozart hardware manufactured today support future larger models?
If we support that by adapting model parallelism on top of your design, will your performance benefits degrade?
And if so, by how much?
Discussing these questions would help address the limitations on hardware scalability.

**Paper Formatting Concerns:**

No format issue observed.

**Quality:**

4

**Strengths And Weaknesses:**

Strength:

1. This paper focuses on popular topics like MoE, wafer-scale, and 3D architecture.

2. This paper proposes a comprehensive solution for placing the experts based on the hardware architecture.

3. The evaluation shows impressive performance benefits.

Weakness:

1. The compute-communication overlap seems similar to existing works like DualPipe from DeepSeek and TileLink from ByteDance.

2. This paper did not discuss that, as model sizes scale quickly, what will happen if the size of an expert exceeds the capacity of one chiplet group. Specifically, when each expert is mapped to multiple chiplet groups, how will you handle the parallelism?

---

> ### Author Rebuttal · Authors · 2025-07-29
>
> We sincerely thank reviewer X4mS for acknowledging our work, especially “enjoy reading this paper and have learned a lot”. To address your concerns, we provide a pointwise response below:
>
> **[Weakness 1 & Questions 1. Communication-computation overlap scheme]**
>
> We appreciate your insightful comments on introducing the relevant research. Although both DualPipe and TileLink are tailored for large-scale GPU clusters, they share some similar design principles with our work, while $\texttt{Mozart}$ also encompasses some specialized mechanisms beyond their scope. We will present a comprehensive comparison, along with elaborating our design innovations as follows:
>
> *Different design principles*
>
> - ***Communication type***: Both TileLink and DualPipe consider overlapping the communication across GPUs, which is typically constrained by the limited bandwidth. Mozart tries to overlap the communication between DRAM and chiplets, which is the dominant bottleneck on wafer-scale chiplet systems.
> - ***Overlapping strategies***: In $\texttt{Mozart}$, the DRAM communication for MoE parameters is overlapped with both the computation for streaming tokens and the hierarchical all-to-all communication across different chiplets, while the DRAM communication for attention parameters is overlapped with the computation for streaming tokens. In TileLink, the cross-device communication is mainly overlapped with tiled computing tasks that are executed alternately. In DualPipe, the cross-device communication is mainly overlapped with computation operations from both fwd and bwd passes (e.g, in fwd pass all-to-all communication can be overlapped with attention computation for other micro batches). Since the architecture hierarchy and performance bottleneck of GPU clusters and our chiplet systems are quite varied, our overlapping strategies are highly specialized for chiplet systems to alleviate the dominant bottleneck.
>
> *Innovations of $\texttt{Mozart}$*
>
> - ***Streaming tokens***: Both DualPipe and TileLink used the concept of micro-batch, while $\texttt{Mozart}$ introduced a similar notion as streaming tokens, which differs from these previous notions. Our motivation is to utilize the heterogeneity of attention and FFN modules on chiplet systems. The streaming tokens are separated for attention and FFN at each layer, isolated with synchronization (as shown in Figure 4). For attention, all the tokens are streamed into evenly distributed micro-batches and processed sequentially. For FFN, the tokens are streamed for each chiplet after the hierarchical all-to-all communication, aiming at overlapping the activation saving communication (between chiplet and DRAM) with expert computation. Different MoE chiplets can be executed in parallel to adhere to the distributed MoE DRAMs.
> - ***Streaming experts***: Different from previous works, Mozart harnesses the profiled routing priors of the pre-trained MoE-LLMs to promote the overlap between off-package DRAM communication and on-package operations, through the proposed streaming experts. Since multiple MoE chiplets share the same DRAM, we load the clustered experts in sequential order following the profiled probability of being routed, which offers an additional benefit for runtime latency.
>
> In a word, our work stems from the fundamental performance bottlenecks of distributed chiplet systems. By harnessing the profiled expert activation & co-activation priors, we effectively enhance the system runtime efficiency in the process of post-training for modern MoE-based LLMs, aiming at providing a generic framework for deployment of all the large-scale sparse models, in the era that sparse scaling has almost become the industrial standard, as the latest frontier models like DeepSeek, Kimi K2, Llama-4, and Qwen3 all possess this architecture. Compared to the pure system-level optimizations, we explore a novel perspective of algorithm-hardware co-design as an early-stage attempt for efficient MoE deployment on domain-specific hardware.
>
> **[Questions 2. Compatibility with distributed training optimizations]**
>
> Model parallelism techniques for large models mainly include pipeline parallel (PP), tensor parallel (TP), and expert parallel (EP). Currently, the trend for scaling LLMs is sparsely scaling the FFN module with MoE, with successful applications on DeepSeek, Qwen3, and Kimi K2. These large models feature the same characteristic, where the attention module only consumes a very small proportion of the total parameters (also demonstrated in Figure 1 on models with smaller scales). We list the potential applications of these model parallelism techniques below:
> - ***PP***: In $\texttt{Mozart}$, we mainly consider the pipelining for attention and MoE within the same layer. While for inter-layer pipelining, $\texttt{Mozart}$ architecture is also compatible by incorporating multiple replicas of individual $\texttt{Mozart}$, and partitioning the integral pipeline evenly across them. The bubble preventing techniques (Zero bubble pipeline parallelism, Qi et al, ICLR 2024) are also compatible in this extended architecture.
> - ***TP***: Tensor parallelism is usually used on embeddings, lm_head, and attention modules. For $\texttt{Mozart}$, these can all be placed on the attention chiplet for computation, loaded in sequential order from the attention DRAM.
> - ***EP***: $\texttt{Mozart}$ mainly targets efficient expert parallelism on chiplet systems. For larger models with higher sparsity degree, we can extend the MoE chiplets, e.g, adopting 16 MoE chiplet clusters if extending routed experts from 64 to 256.
>
> In a word, $\texttt{Mozart}$ architecture is compatible and scalable for larger-scale distributed training of MoE-LLMs through the integration of multiple model parallelism techniques.
>
> **[Weakness 2 & Limitations. Scalability of $\texttt{Mozart}$]**
>
> Thank you for the insightful question. As demonstrated in our experiments, the current $\texttt{Mozart}$ architecture is capable of supporting the typical expert sizes found in mainstream MoE models, meeting the demands of most practical applications today. To address the challenges brought by future model scaling, we envision two main directions for extending support.
> - ***Increasing the number of experts***: When model scaling is achieved by adding more experts, we can expand the number of switch groups at the hardware level to accommodate more expert chiplets. On the algorithmic side, we can also cluster or merge infrequently activated experts to limit the total number of experts, thereby avoiding communication congestion and maintaining efficient scheduling.
> - ***Growing the size of a single expert***: When an individual expert’s parameter size exceeds the capacity of a single chiplet group, we can adopt intra-expert model parallelism by partitioning the expert’s MLP module across multiple chiplet groups using tensor parallelism locally. Since $\texttt{Mozart}$ adopts a hierarchical NoP-Tree topology, where switch chiplets natively support in-network aggregation and scheduling (e.g., reduce-scatter and all-gather operations), this architecture is well-suited for executing experts in parallel across chiplets. In summary, while $\texttt{Mozart}$’s current design assumes a one-expert-per-group configuration, the underlying NoP-Tree interconnect and communication mechanisms are inherently scalable, making the architecture well-equipped to support expert scaling with tremendous sparsity and the continued evolution of large-scale MoE-based large language models.

---

> > ### Comment · Reviewer_X4mS · 2025-08-01
> >
> > Thanks for the response!
> >
> > Q1: Then the question becomes, is it possible to adapt their overlap strategy to Wafer? For example, following DualPipe, we can separate one inference batch into two microbatches A and B, then we can execute DRAM A & compute B, compute A & DRAM B, and repeat, in an interleaved manner. Then, why is your approach preferred over the above one?
> >
> > Q2&3: Solved.

---

> > > ### Author Response · Authors · 2025-08-02
> > >
> > > Dear reviewer X4mS:
> > >
> > > Thanks for raising such an intriguing suggestion; however, a DualPipe-style overlapping strategy is not feasible on Mozart. The reason lies in 2 aspects:
> > >
> > > 1. The dominant bottleneck of Mozart is the DRAM communication overhead for MoE modules within a layer (please refer to figures 1 and 4). Although the execution duration of MoE can be utilized for other attention micro-batches, the attention duration can hardly accommodate other MoE micro-batches, since the parameter loading process has already occupied such an interval.
> > > 2. We have already adopted a similar micro-batch splitting strategy in Mozart (please refer to Figure 4 and Section 4.3 for the scheduling algorithms). However, since Mozart employs an *Attn-FFN* disaggregation strategy, the batching algorithms for the two modules differ significantly. Additionally, batching on MoE chiplets must account for the dispatching results (expert activation patterns). Introducing an additional DualPipe-style micro batch splitting strategy can be not only unnecessary but also complex to implement.
> > >
> > > Therefore, although the Mozart pipeline admits minor optimizations, the available space is far too limited to accommodate the overhead and complexity of a DualPipe-style overlapping scheme with supplemental micro-batches.

---

> > > > ### Comment · Reviewer_X4mS · 2025-08-02
> > > >
> > > > Thanks for your reply!
> > > > That addresses my question.

---

### Official Review · Reviewer_6v38 · 2025-07-04

**Clarity:** 3
**Significance:** 2
**Originality:** 3
**Rating:** 3
**Confidence:** 5

**Summary:**

This paper presents a 3.5-D hardware architecture that leverages new in-package semiconductor technologies to tightly integrate small chiplets at a large scale, connected together using a hierarchical network on-chip. It is labeled as 3.5-D (instead of 2.5-D) because each chiplet has 3D integration of SRAM (memory) on top of a logic die as well. On the algorithmic side, the paper presents a number of architecture-specific optimizations, specifically to optimize the post-training fine-tuning of mixture-of-expert architectures such as Deepseek, Olmo, and others. The proposed optimizations are: (1) clustering experts and pins them to the suitable chiplet in the system hierarchy to minimize overall communication, and (2) pipelines memory access to hide computation latency.

**Questions:**

The main missing component of this paper is to compare to any existing hardware solution to enable some understanding of the impact of this work. Are you able to compare, for example, to 8xA100?

**Ethical Concerns:**

["NO or VERY MINOR ethics concerns only"]

**Final Justification:**

Additional comparisons to GPUs, while rough, strengthen the paper. However, I am not totally convinced of the evaluation and methodology yet.

**Quality:**

2

**Strengths And Weaknesses:**

Strengths: the presented architecture is very interesting and combines many new ideas to tackle large-model fine-tuning. Furthermore, there are RTL simulated areas with details down to the pitch of TSVs suggesting a realistic hardware architecture. Although the focus of the paper is on the mapping of MOE model mapping onto the device. On that front, the presented ideas are interesting and logical, but:

Weaknesses:
1- There are no comparisons, even high-level, rough comparisons to any available hardware. This makes it very difficult to appreciate the presented hardware architecture, its uitility, or efficiency compared to available solutions such as GPUs.
2- Validation: Barring tape-out, it is conventional in the hardware space to map a small design onto an FPGA or a cycle-accurate simulator to validate the cycle-accurate assumptions made in the paper.
3- MOE mapping methods are somewhat straightforward. If I assume the novelty of the paper is on the hardware side, I find that this is not the right venue for presentation, if the novelty is in specifically mapping MOE to this new hardware architecture, I find that the presented methods are practically sound and logical but somewhat straightforward.

---

> ### Author Rebuttal · Authors · 2025-07-28
>
> We thank reviewer 6v38 for acknowledging our work, which “is interesting and logical, combines many new ideas”. We supplement the omitted runtime comparison and the novelty of mapping to address your concerns.
>
> **[Weakness 1 & Questions. Comparison with A100]**
>
> Thanks for this constructive suggestion. We conducted comparisons on an advanced server with 8x NVIDIA A100 GPUs (each with 80GB HBM) to mimic $\texttt{Mozart}$, which features 4 chiplet clusters and has a similar total power consumption. Any two GPUs are interconnected with 12 NVLink in this server, providing a bidirectional bandwidth of around 400 GB/s (as verified through peer-to-peer communication using send & recv in torch.distributed). The CPU and memory configurations of this server are:
>
> |CPU Config|Values|
> |-|-|
> |CPU(s):|152|
> |On-line CPU(s) list:|0-151|
> |Model name:|Intel(R) Xeon(R) Platinum 8368 CPU @ 2.40GHz|
> |Thread(s) per core:|2|
> |Core(s) per socket:|38|
> |Socket(s):|2|
> |CPU(s) scaling MHz:|31%|
> |NUMA node0 CPU(s):|0-37,76-113|
> |NUMA node1 CPU(s):|38-75,114-151|
>
> |Memory Config|total|used|free|shared|buff/cache|available|
> |-|-|-|-|-|-|-|
> |Mem:|1.0Ti|158Gi|835Gi|33Gi|53Gi|849Gi|
> |Swap:|9Gi|4.2Gi|5.8Gi|
>
> We adopt PyTorch’s Fully Sharded Data Parallel (FSDP)—the $de facto$ standard for fine-tuning open-source LLMs on the Hugging Face ecosystem—as our GPU baseline, leveraging its ZeRO-style parameter sharding and optimization strategies for advanced runtime efficiency and scalability. We set the local batch size to 4 in GPU baselines to keep consistent with the global batch size in $\texttt{Mozart}$. We also report the average training latency per step for 1k iterations. We examine both offloading and non-offloading configurations, which are both commonly employed techniques when fine-tuning large models with FSDP. We utilize FlashAttention 2 to accelerate the attention module in all experiments, thereby minimizing the influence of attention computation overhead.
>
> *Sequence Length 128*
> |Model|Qwen3|DeepSeek|OLMoE|
> |-|-|-|-|
> |$\texttt{Mozart}$ (HBM2)|2.65|0.57|1.48|
> |PyTorch FSDP|6.16 (2.32x)|1.14 (2x)|2.48 (1.68x)|
> |PyTorch FSDP (Offload)|28.36 (10.70x)|5.75 (10.09x)|13.39 (9.05x)|
>
> *Sequence Length 256*
> |Model|Qwen3|OLMoE|DeepSeek|
> |-|-|-|-|
> |Mozart (HBM 2)|2.75|0.77|1.56|
> |PyTorch FSDP|6.42 (2.33x)|1.23 (1.60x)|2.65 (1.70x)|
> |PyTorch FSDP (Offload)|28.70 (10.44x)|5.85 (7.60x)|13.78 (8.83x)|
>
> *Sequence Length 512*
> |Model|Qwen3|OLMoE|DeepSeek|
> |-|-|-|-|
> |Mozart (HBM 2)|3.26|1.04|2.46|
> |PyTorch FSDP|6.80 (2.09x)|1.73 (1.66x)|3.20 (1.30x)|
> |PyTorch FSDP (Offload)|29.64 (9.09x)|6.26 (6.02x)|14.21 (5.78x)|
>
> Compared to the GPU baselines, $\texttt{Mozart}$ mainly features the following properties:
> - ***Improved efficiency***: $\texttt{Mozart}$ consistently surpasses GPU counterparts across all the baselines with a significant reduction in training latency.
> - ***Better scalability***: $\texttt{Mozart}$ demonstrates a higher speedup with Qwen3-30B-A3B model on both offloading and non-offloading settings, which contains much larger parameters than OLMoE-1B-7B and DeepSeek-MoE-16B.
>
> **[Weakness 2. Cycle-accurate simulator]**
>
> Thank you for the valuable feedback. While we have not yet mapped the full Mozart system onto an FPGA or performed tape-out, we have implemented the logic chiplets, SRAM chiplets, inter-chiplet interconnects, and switches in Verilog and synthesized them with Synopsys Design Compiler under 28nm technology. Furthermore, we have developed a cycle-level simulator, whose key components have been validated against Verilog simulation results in terms of runtime and power to ensure accuracy. Therefore, our simulation methodology provides high fidelity, following the common practices.  As part of our future work, we plan to perform a full-system FPGA validation of Mozart and eventually proceed to tape-out. For simulation details, please refer to Section 5.2 of our paper.
>
> **[Weakness 3. Novelty]**
>
> We respectfully disagree with the assessment that the contribution is either “hardware-only” or “straightforward.” Our paper focuses on algorithm-hardware co-design rather than pure hardware or algorithm innovations. Therefore, our innovations lie in 3 aspects, including hardware-side, algorithm-side, and co-design-side, listed as follows:
>
> *Hardware-side:*
>
> The hardware innovations of $\texttt{Mozart}$ are proposed to tackle the heterogeneity of attention and FFN modules, along with the heavy communication overhead in MoE-LLMs. The insights are listed as follows:
> - ***3.5D vertical logic-on-memory stack***: To improve memory locality, we design a hierarchical memory system aligned with the temporal reuse patterns of MoE-LLMs, enabling frequently reused data to be cached closer to the computing unit using the 3D logic-on-memory stack, thereby reducing costly accesses to off-chip DRAM;
> - ***2.5D NoP-Tree topology with switches & DRAMs***: Mozart adopts a 2.5D NoP-Tree interconnect where attention chiplets sit at the root and MoE chiplets form the leaves. Each group of four MoE chiplets shares a DRAM channel and connects through a switch chiplet with local aggregation support, reducing inter-group bandwidth contention. By co-locating frequently activated experts on the same chiplet and applying scheduling-aware dataflow, the system minimizes memory and communication bottlenecks to improve throughput.
>
> *Algorithm-side:*
>
> Our algorithm-side innovations focus on expert-chiplet mapping and fine-grained streaming schedules. The insights of $\texttt{Mozart}$ are listed as:
> - ***Advancing communication efficiency with expert activation & collaboration awareness***: Our expert-chiplet mapping scheme does not simply follow conventional expert parallelism, i.e, evenly allocating experts on chiplets without considering the profiled routing priors. As we pointed out in Section 3.3, expert placement can affect the all-to-all communication volume, which is a critical bottleneck in expert parallelism. Therefore, we develop novel expert clustering & allocation algorithms to specialize in grouped expert layout for communication efficiency, which is demonstrated as $\texttt{Mozart-B}$ & $\texttt{Mozart-C}$ in Figure 6.
> - ***Streaming schedules for improved parallelism***: Our fine-grained streaming schedules stem from the heavy DRAM communication volume in MoE deployment, as well as the all-to-all communication overhead. We designed streaming tokens & experts as fine-grained scheduling algorithms to overlap the high communication budgets. The effectiveness is demonstrated in $\texttt{Baseline}$ & $\texttt{Mozart-A}$ in Figure 6, where the performance improvement can be more significant under longer input sequences.
>
> *Co-design-side:*
>
> To efficiently deploy MoE-LLMs, we proposed more tightly-coupled designs to integrate the efficiency-oriented algorithm innovations and the sparsity-driven hardware hierarchy. The insights are listed as follows:
> - ***Expert allocation algorithm & NoP-tree topology***: We proposed a 2.5D NoP-tree interconnect specialized for the MoE deployment. The motivation lies in tackling the heterogeneity of the Attention and FFN modules with our proposed grouped expert-chiplet assignment for communication efficiency. We validate the effectiveness in Table 4, where our coupled design can reduce both all-to-all communication volume and latency.
> - ***Streaming schedule algorithm & Memory hierarchy***: We assign a DRAM chip for both the attention chiplet and MoE chiplet groups to alleviate the heavy burden of DRAM communication through specialized storage, and employ streaming algorithms to overlap off-package communications with on-package computations fully.

---

> > ### Author Response · Authors · 2025-08-03
> > **Request for Further Discussion with Reviewer 6v38**
> >
> > Dear Reviewer 6v38,
> >
> >
> > We sincerely thank Reviewer 6v38 once more for your thoughtful and constructive feedback. We have carefully addressed every concern in our response and would greatly value your perspective on whether the clarifications and additional experiments fully resolve the issues you highlighted.
> >
> >
> > We genuinely hope Reviewer 6v38 could kindly check our response. Thank you!
> >
> >
> > Best wishes,
> >
> >
> > Authors

---

> > > ### Author Response · Authors · 2025-08-04
> > > **We are keen to discuss further with Reviewer 6v38**
> > >
> > > Dear Reviewer **6v38**,
> > >
> > >
> > > Thank you for your valuable time and the constructive feedback you have provided once again. We are eager to engage in further discussions to see if our response solves your concerns.
> > >
> > >
> > > As the **deadline** for the discussion period is approaching, we would greatly appreciate it if you could kindly let us know if you have any further questions. Thank you again for your attention to our work.
> > >
> > >
> > > Best wishes,
> > >
> > >
> > > Authors

---

### Official Review · Reviewer_Bfvg · 2025-07-07

**Clarity:** 3
**Significance:** 3
**Originality:** 4
**Rating:** 5
**Confidence:** 5

**Summary:**

This paper introduces Mozart, a software-hardware co-design framework for the efficient post-training of MOE LLMs. This work is motivated by the hardware challenges posed by the sparse and modular nature of MoE models especially memory locality and communication overhead. The proposed strategy tries to addresses expert routing and communication overheads in MoE-style LLMs, On the algorithmic side, Mozart implements an expert allocation strategy that clusters experts based on their co-activation patterns to minimize inter-chiplet communication and balance workloads. It also implements a fine-grained scheduling mechanism to improve the overlap between communication and computation. On the hardware side, Mozart designs a specialized 3.5D wafer-scale chiplet architecture with a hierarchical memory system and a 2.5D NoP-Tree interconnect topology. Using cycle-accurate simulations, the paper shows that Mozart achieves speedups (over 1.9x) across three popular MoE models compared to baseline designs.

Despite its specialization to MoE-style LLMs and modular hardware assumptions, the paper presents valuable design principles for future accelerators. Its latency-aware expert scheduling, communication modeling, and hardware-algorithm co-design offer practical insights for chiplet-based and sparse-aware systems. These contributions are particularly timely as AI hardware trends shift toward modular packaging, inter-die communication optimization, and efficient support for dynamic, sparse workloads. While generalization beyond MoEs is limited, the paper nonetheless informs a growing class of AI accelerators targeting scalable, fine-grained model execution.

**Questions:**

1.	The expert clustering approach (Algorithm 1) is based on a greedy, farthest-point-sampling method. Have you investigated or considered alternative graph clustering algorithms (e.g., spectral clustering, METIS) on the expert co-activation graph especially that the used algorithm is computationally intensive? It would be good to know if other methods could yield a better workload balance or further reduce communication volume.
2.	Can you elaborate on how the system's performance and communication patterns would be affected if you were to apply data or tensor parallelism across multiple chiplets for the attention computation? How would this interact with the NoP-Tree topology? Would it be compatible with existing TP approaches?
3.	The expert allocation strategy is based on a static analysis of expert co-activation patterns from an instruction tuning dataset. How stable are these patterns during a long pre-training run? Would a dynamic approach, where expert clusters are periodically re-evaluated and re-allocated, be necessary or beneficial?
4.	All-to-all communication is identified as a key bottleneck. Your approach reduces the data volume by co-locating experts. How does this compare to other techniques for optimizing all-to-all, such as using hierarchical communication algorithms or topology-aware communication libraries?
5.	Your proposed hardware–software co-design is tightly coupled to the modular and sparse nature of MoE architectures. I'm not expecting a complete solution here, but could you share how you might begin to extend or adapt your approach to support a broader range of model types—such as dense LLMs, hybrid MoE + dense architectures, or even non-transformer models like state-space models (e.g., Mamba)? In particular, how might your expert clustering, routing strategies, or inter-chiplet communication optimizations evolve (or be abstracted) to support models with different compute and dataflow characteristics?

**Ethical Concerns:**

["NO or VERY MINOR ethics concerns only"]

**Limitations:**

yes

**Quality:**

3

**Strengths And Weaknesses:**

Strengths:
- The paper is well written and organized.
- The authors address a timely topic.  MoE architectures are increasingly being adopted to scale LLMs to trillion-parameters. Hence addressing their communication overhead and resource utilization bottleneck is an important and practical research direction.
- The concept of an algorithm-hardware co-design frameworks specifically designed for MoE models on emerging 3.5D wafer-scale architectures is a strong contribution.
- The work is well motivated by a clear analysis of the system-level challenges.
- The of a cycle-accurate simulator that has been validated against Verilog simulations is also a strong point.
- The emprirical evaluation is generally well explained and done.

Weaknesses/Areas of improvements:
- The hardware evaluation is entirely simulation-based. While this is a critical first step in any new architecture research, the performance evaluations and gains depend heavily on the realization of the 3.5D wafer-scale system. I recommend that the authors include a detailed discussion about the assumptions made in the simulator and if there are any challenges to translate their design and simulation into a real physical hardware/chip.
- This statement is partially correct: “Prior works tend to neglect wafer-scale integration [7, 19, 42] and largely adopt coarse-grained, static workload partitioning strategies [29].”. Wafer-scale integration (e.g., Cerebras WSE) is rare and only done by a few companies due to extreme cost, yield, and packaging challenges. So although most existing work do not consider wafer-scal, this is rather a of practicality than oversight. Coarse-grained, static partitioning is common in chiplet designs: models are statically sharded across chiplets based on layers or tensor slices. That said, some recent works that are still evolving (e.g., FlexGen, Megatron-DeepSpeed) do explore dynamic routing or load-balancing. ”
- The statement in the paper: “Prior works lack system-level coordination and optimization for MoE” uses reference 22, Future of Memory (Liu et al., IEDM 2024). This reference is not a MoE-LLM workload partitioning paper strategy. This work advocates for tighter integration of diverse memory systems but does not propose specific coordination or workload tiling strategies for MoE-LLMs or chiplet systems. So, this citation does not directly support the claim about workload tiling or coordination strategies for MoE.
- Some key relevant works are missing from the related work discussion:
   - GShard (Lepikhin et al., ICLR 2021): A foundational MoE system that introduced routing + automatic sharding across devices. Crucial for historical context. This is the first large-scale MoE at Google.
   - BASE Layers (Lewkowycz et al., NeurIPS 2022): A hybrid sparse-dense expert design that maintains performance while reducing memory. This should be compared and contrasted Mozart’s memory-aware scheduling.
    - MoEfication (Zhou et al., ICML 2022): Shows conversion of dense models to MoE post-training. This is related to the post-training MoE optimization approaches as proposed by Mozart.
   - TaskMoE / Expert Tuning (Zhang et al., 2022): Specialization for multi-task models.
   - Tutel & DeepSpeed-MoE need to be more discussed, especially in comparison to Mozart’s routing, scheduling, and clustering.
- Suggestions for improving the related work section:
   - Cite GShard and discuss how Mozart differs in routing granularity, post-training design, and hardware assumptions.
   - Include Cerebras WSE to contrast wafer-scale monolithic vs. modular chiplet-based wafer-scale.
   - Add a table or paragraph contrasting Mozart with FRED, Cambricon-LLM, Maestro, etc., clarifying MoE-awareness and system-level scheduling gaps.
   - Consider adding a  short discussion of commercial accelerators (AMD MI300, Intel PV) for practical relevance.
   - Cite MoEfication to better support the post-training and fine-grained modularity motivation.
- The clustering method used in Equation 5 is based on the k-clustering with pairwise distances.This algorithm is computationally expensive (O(n²)), and no mention is made of approximation methods (e.g., hierarchical clustering, METIS).
- The mathematical formulation in Equation 6 does not provide constraints on the number of experts per token (e.g., top-2) of the token capacity per expert (e.g., max tokens per batch). This needs to be clarified by the authors.
- In Equation 7, runtime score for scheduling, how is the load defined? Is it measured in flops, tokens, latency? What is the utilization measured against? (e.g., theoretical peak). The formulation remains abstract and hard to reproduce without further clarification by the authors.
- The proposed co-design strategy effectively addresses expert routing and communication overheads in MoE-style LLMs, however, it is highly specialized to this sparse model family. The system assumes hard top-k expert gating and token-level dispatch, which do not generalize to dense autoregressive models (e.g., GPT-3, LLaMA), diffusion models, or state-space architectures (e.g., Mamba) that lack modular sparse execution patterns. Additionally, the hardware assumptions—modular chiplet-based designs with a Network-on-Package interconnect—limit applicability to a narrow class of accelerators. Real-world systems such as monolithic GPUs, wafer-scale engines (e.g., Cerebras), and mobile AI chips follow vastly different architectural and memory paradigms. The lack of runtime adaptability in expert scheduling and coarse treatment of memory hierarchies further constrain its deployability. A discussion on these generalization limitations is absent.
- Regarding the empirical evaluation. There is a lack of baselines beyong static scheduling. The main baseline is a static MoE routing strategy or uniform tiling. It would be good to add comparisons against DeepSpeed-MoE or Tutel routing strategies and Existing expert placement heuristics (e.g., token-balanced sharding, load-aware dispatch). The existing evaluation is not enough to claim that Mozart significantly outperforms state-of-the-art MoE runtimes.
- Limited System Scaling Study. The paper does not analyze how Mozart behaves under increasing number of chiplets, varying interconnect topologies (e.g., mesh vs. ring) and realistic memory capacity constraints. Such scaling or sensitivity analysis can showcase if the benefits hold at larger or smaller deployment scales.
- The paper’s results report mean latency or throughput, but does not show variance or standard deviation, confidence intervals, and sensitivity to input token distribution (e.g., random vs. clustered token-to-expert assignment).
- While latency and utilization are discussed, the paper completely omits energy or area modeling. Given the focus on chiplet hardware, this is a notable omission—MoE communication patterns can be power-intensive.

---

> ### Author Rebuttal · Authors · 2025-07-28
>
> We sincerely thank reviewer Bfvg for the insightful and detailed comments. To address your questions, we provide pointwise responses below.
>
> **[Weakness 1. Assumptions of the simulator]**
>
> Mozart is simulated on a tree-structured chiplet system with 27 heterogeneous chiplets: 1 for attn, 16 for FFN experts (4 clusters x 4 chiplets), 4 for switches, and 6 for DRAMs, organized in a balanced quadtree of depth 2.
>
> Each compute chiplet employs 3D hybrid bonding, stacking high-density distributed SRAM beneath the compute logic to provide significantly higher bandwidth and capacity than conventional 2D integration. Inspired by wafer-scale systems like Cerebras, Mozart offers sufficient on-chip SRAM to hold expert parameters and pre-activated activations, eliminating intra-chiplet caching bottlenecks. For area efficiency, expert chiplets adopt a weight-streaming approach: only the parameters of the activated experts are prefetched from the local cluster DRAM in each training cycle. The top logic layer integrates weight-stationary PEs for GEMM operations and standard non-linear functions.
>
> Each inter-chiplet link provides 256 GB/s bandwidth and incurs a 1 ns per-hop latency. Switch chiplets support in-network computing, enabling efficient collective operations (e.g., token scatter/gather and gradient aggregation) required for fwd and bwd passes. Our simulation models both computation and inter-chiplet NoC routing at cycle level to evaluate token latency under warm-up and steady-state conditions.
>
> **[Weakness 2. Partially correct statement]**
>
> We will refine it following your constructive comments.
>
> **[Weakness 3. Incorrect citation on Future of Memory]**
>
> We apologize for the omission, and will replace [22] with Cambricon-LLM (Yu et al, ISCA 2024) and Hecaton (Huang et al, arxiv 2024).
>
> **[Weakness 4 & 5. Related works and discussions]**
>
> We will add a subsection after *Modularized LLMs* to incorporate these fundamental works on GPU-deployment of MoE-LLM.
>
> **[Weakness 6. Computation complexity of expert clustering]**
>
> The clustering problem is solved offline on the profiled data. Since all the latest MoE-LLMs (even the largest Kimi K2) use only a handful of routed experts (i.e, 384) per layer, the expert assignment problem is compact enough to be solved exactly with commercial solvers such as Gurobi.
>
> **[Weakness 7. TopK]**
>
> We follow the dropless MoE design in MegaBlocks without the notion of expert capacity. We will claim it in our revised version.
>
> **[Weakness 8. Runtime score]**
>
> We use Equation (7) to demonstrate the effectiveness of measuring all-to-all communication volume with our proposed $C_{\mathcal{T}}$. The workload here is the tokens before MoE computing. It is not a runtime scheduling scheme, which lies in Section 4.3.
>
> **[Weakness 9 & Question 5. Generalization]**
>
> Mozart targets deploying MoE-LLMs on chiplet systems, as sparse scaling has been widely adopted in the latest LLMs (e.g, DeepSeek, Qwen3, and Kimi K2), but studies on specialized hardware are very few.
>
> From the hardware side, modularization enables flexible composition of heterogeneous components on chiplet systems for diverse applications, which is a key trend in AI accelerator design. This modular design is also validated in practice by a 3D SoC partition and composition Compiler platform from Synopsys, which has been adopted by industry to accelerate AI system development. Mozart adopts a modular design with 2 specialized chiplets for memory-bound attn and compute-bound FFN, respectively, enabling flexible workload-to-chiplet mapping by decoupling operators based on their characteristics.
>
> Though tailored for MoE, Mozart is composable and generalizable. Dense models are also FFN-dominant, and overlapping DRAM communication is also critical. FFN can be decomposed into parallel subtasks mapped to multiple chiplets. We can remove the routing designs, evenly allocate FFN shards on previous MoE chiplets, and replace all-to-all communication with scatter & gather. Although the NoP-Tree in Mozart is physically implemented at the chiplet level, its concept can serve as a logical topology in GPU or wafer-scale systems. For example, GPUs can emulate tree-based expert routing in software using NCCL, while Cerebras WSE, as a wafer-scale monolithic, allows logical NoP-Tree topology via programmable routing. This aligns with FRED (ISCA 2025), which advocates tree-like interconnects over rigid 2D meshes to better support flexible, non-blocking collective communication.
>
> Overall, although Mozart is optimized for MoE deployment, its core principles, modular compute decomposition, communication-aware scheduling, and logical topology design, are highly extensible.
>
> **[Weakness 10. Baselines]**
>
> We omitted to claim that we follow dropless MoE in MegaBlocks to enable seamless adaptation to pre-trained MoE-LLMs. Different routing strategies or expert layout heuristics for load balancing can be easily integrated into Mozart.
>
> **[Weakness 11. System scaling]**
>
> Mozart is an empirical optimal solution for MoE-LLMs because:
> - Mozart supports moderate scaling by expanding expert chiplet clusters and switch nodes. However, as the number of MoE chiplets grows excessively, token replication and attn-to-expert communication workloads also grow, leading to higher all-to-all burdens. Furthermore, assigning a certain number of additional chiplets to the attn module can reduce its computation latency; however, this introduces extra routing complexity within the attention group. Therefore, a trade-off must be made between computing latency and routing overhead to maintain system efficiency.
> - We apply in-network computing switches to establish inter-chiplet connections. The tree topology with such switches is inherently well suited for essential reduce/scatter operations in MoE workloads, while sacrificing extra area as the switch chiplet is nearly as large as a computing die.
> - We apply mesh to establish intra-chiplet connections, since it is feasible for regular data movement in GEMM.
>
> **[Weakness 12. Comprehensive results]**
>
> For more numerical results, we will investigate factors that potentially affect simulation performance and update the results & visualizations. For sensitivity to input token distribution, we will examine more post-training datasets from multiple domains under different expert layouts in our revised version.
>
> **[Weakness 13. Energy & Area modeling]**
>
> |Qwen3-30B-A3B|Area (mm^2)|Energy/Step (J)|
> |-|-|-|
> |Attn|474|2842|
> |MoE|6562|3470|
> |Switches & Interconnect|5839|1320|
> |DRAM|1300|1541|
>
> Attn module has a smaller footprint than interconnect or MoE chiplets due to per-chiplet area constraints, limiting local caching & computational capacity and leading to more frequent intra-chiplet data movement and higher energy consumption regarding its area. Moreover, most energy is consumed by computing modules rather than DRAM or interconnect, confirming the effectiveness of our communication optimization, as switch energy remains low despite large-scale sparse expert activation.
>
> **[Questions 1. Expert Clustering]**
>
> Since profiling and expert assignment are conducted offline before deployment, the efficiency of these algorithms is not very critical; thus, we can try to solve the exact results for the binary programming task. We will try to solve the expert allocation task using alternative formulations in our future work.
>
> **[Question 2. DP/TP for Attention Layer]**
>
> Though DP and TP are not considered in Mozart (stated in "Conclusion and Limitations"), they can be easily adopted when attn requires more resources beyond an individual chiplet with area constraints. First, the NoP-Tree topology with hierarchical routing and in-network aggregation at switch chiplets naturally supports parallel patterns such as all-gather & reduce-scatter. Distributing attention heads (TP) or input tokens (DP) across multiple smaller attention chiplets would increase area usage and introduce additional inter-chiplet communication overhead. However, the increased computing resources may help reduce computing latency. The overall performance would reflect a trade-off among factors such as per-die area, the number of sub-chiplets, and routing complexity. Second, our chiplet partition is compatible with existing TP frameworks like Megatron-LM, as they typically partition the weight tensors within each model layer and perform frequent all-gather and reduce-scatter operations. Mozart’s NoP-Tree topology with in-network computing switches natively supports such communication primitives, offering efficient collective capabilities. Conversely, the mesh structure we employed in the single computing die is compatible with GEMM and GEMV computations with ring or torus logical topology. The dual-topology design enables Mozart to optimize both intra-chiplet computation and inter-chiplet communication overhead.
>
> **[Question 3. Expert allocation]**
>
> Since Mozart only targets post-training, it is rational to assume that the collaboration pattern would not change much during tuning. But for pre-training, the expert layout at each layer should be dynamically adjusted to continuously benefit from the efficient all-to-all communication it brings. It is complex, as it requires recording routing choices continuously, fast expert allocation algorithms, and dynamic expert layout updating, which is beyond the scope of this paper. We will try to make a more systematic resolution with algorithm-system co-design as our future work.
>
> **[Question 4. All-to-all communication]**
>
> Mozart has conducted these 2 aspects, including hierarchical routing (first determine the routed chiplet cluster, then determine the specific chiplet) and topology-aware communication (the 2.5D NoP-tree topology) to accommodate this refactored routing. It can be seamlessly applied to any MoE-LLMs as it does not modify the routing policy of the pretrained model.
>
> **[Question 5. Extension to more LLMs]**
>
> Please refer to *Weakness 9*.

---

### Note · Authors · 2025-08-12

Dear **ACs** and **Reviewers**,

In this paper, we propose Mozart, a novel 3.5D wafer-scale chiplet system enabling modularized and efficient training for MoE-LLMs. We integrate novel techniques to achieve efficient sparse training, including specialized expert layout for efficient all-to-all communication, fine-grained scheduling to overlap DRAM communication, and a wafer-scale chiplet system for MoE architecture.

We are grateful for the reviewers' recognition, including comments like ‘enjoy reading this paper and have learned a lot’, ‘well written and organized’, and ‘address a timely topic’, and appreciation for our technical innovations, such as ‘the presented architecture is very interesting and combines many new ideas to tackle large-model fine-tuning’, and ‘RTL simulated areas with details down to the pitch of TSVs suggesting a realistic hardware architecture’.

During the rebuttal period, we addressed these key concerns:
- **1. Comparison with available hardware** (Reviewer **6v38**): We provide additional experimental results to compare Mozart with a single 8×A100 node running a widely used LLM-tuning framework, revealing that Mozart consistently achieves lower training latency, demonstrating its superiority.
- **2. Innovation of communication-computation overlap** (Reviewer **X4mS**): We clarified the innovation of our communication-computation overlap scheme against existing works by declaring the distinct dominant bottleneck in chiplet systems, and the insights of our scheduling algorithms implemented with streaming tokens and experts. We also validated the effectiveness in the experiments.
- **3.Generalization to more architectures** (Reviewer **Bfvg**): We demonstrate that Mozart’s design principles can generalize well beyond MoE models: with only modest reconfiguration of the hardware, it can seamlessly extend to dense transformer variants.
- **4.Scalability to larger models** (Reviewer **X4mS**): We discussed deploying larger MoE models on Mozart from the hardware side, revealing that our novel NoP-Tree topology and the 3.5D chiplet architecture are natively scalable for large-scale sparse models.

The constructive feedback has greatly helped us strengthen our work. We will ensure the final version incorporates these expanded discussions and experiments.

We deeply appreciate the ACs' and reviewers' valuable time and constructive feedback, which greatly contributed to strengthening the quality and clarity of our paper.

Best,

Authors

---

### Decision · Program_Chairs · 2025-09-17

**Decision:**

Accept (spotlight)

**Comment:**

(a) Summarize the scientific claims and findings of the paper based on your own reading and characterizations from the reviewers.

This paper introduces Mozart, a software-hardware co-design framework for the efficient post-training of Mixture of Experts (MoE) LLMs. This work is motivated by the hardware challenges posed by the sparse and modular nature of MoE models especially memory locality and communication overhead. The proposed strategy tries to addresses expert routing and communication overheads in MoE-style LLMs, On the algorithmic side, Mozart implements an expert allocation strategy that clusters experts based on their co-activation patterns to minimize inter-chiplet communication and balance workloads. It also implements a fine-grained scheduling mechanism to improve the overlap between communication and computation. On the hardware side, Mozart designs a specialized 3.5D wafer-scale chiplet architecture with a hierarchical memory system and a 2.5D NoP-Tree interconnect topology.  It is labeled as 3.5-D (instead of 2.5-D) because each chiplet has 3D integration of SRAM (memory) on top of a logic die as well. On the algorithmic side, the paper presents a number of architecture-specific optimizations, specifically to optimize the post-training fine-tuning of mixture-of-expert architectures such as Deepseek, Olmo, and others. The proposed optimizations are: (1) clustering experts and pins them to the suitable chiplet in the system hierarchy to minimize overall communication, and (2) pipelines memory access to hide computation latency. Using cycle-accurate simulations, the paper shows that Mozart achieves speedups (over 1.9x) across three popular MoE models compared to baseline designs.
Despite its specialization to MoE-style LLMs and modular hardware assumptions, the paper presents valuable design principles for future accelerators. Its latency-aware expert scheduling, communication modeling, and hardware-algorithm co-design offer practical insights for chiplet-based and sparse-aware systems. These contributions are particularly timely as AI hardware trends shift toward modular packaging, inter-die communication optimization, and efficient support for dynamic, sparse workloads. While generalization beyond MoEs is limited, the paper nonetheless informs a growing class of AI accelerators targeting scalable, fine-grained model execution.

(b) What are the strengths of the paper?

The paper is well written and organized.
The authors address a timely topic. MoE architectures are increasingly being adopted to scale LLMs to trillion-parameters. Hence addressing their communication overhead and resource utilization bottleneck is an important and practical research direction.
The concept of an algorithm-hardware co-design frameworks specifically designed for MoE models on emerging 3.5D wafer-scale architectures is a strong contribution.
The work is well motivated by a clear analysis of the system-level challenges.
The of a cycle-accurate RTL simulator, with details down to the pitch of TSVs suggesting a realistic hardware architecture, that has been validated against Verilog simulations is also a strong point.
The empirical evaluation is generally well explained and done.

(c) What are the weaknesses of the paper? What might be missing in the submission?

The hardware evaluation is entirely simulation-based. While this is a critical first step in any new architecture research, the performance evaluations and gains depend heavily on the realization of the 3.5D wafer-scale system. However, this is a minor weakness at this point in the system's development. It is acceptable to have cycle-accurate simulations at this stage, as these provide significant evidence for the ultimate successful deployment of the physical hardware.

Some key relevant works are missing from the related work discussion:
GShard (Lepikhin et al., ICLR 2021): A foundational MoE system that introduced routing + automatic sharding across devices. Crucial for historical context. This is the first large-scale MoE at Google.
BASE Layers (Lewkowycz et al., NeurIPS 2022): A hybrid sparse-dense expert design that maintains performance while reducing memory. This should be compared and contrasted Mozart’s memory-aware scheduling.
MoEfication (Zhou et al., ICML 2022): Shows conversion of dense models to MoE post-training. This is related to the post-training MoE optimization approaches as proposed by Mozart.
TaskMoE / Expert Tuning (Zhang et al., 2022): Specialization for multi-task models.
Tutel & DeepSpeed-MoE need to be more discussed, especially in comparison to Mozart’s routing, scheduling, and clustering.

The clustering method used in Equation 5 is based on the k-clustering with pairwise distances. This algorithm is computationally expensive (O(n²)), and no mention is made of approximation methods (e.g., hierarchical clustering, METIS).
The mathematical formulation in Equation 6 does not provide constraints on the number of experts per token (e.g., top-2) of the token capacity per expert (e.g., max tokens per batch). This needs to be clarified by the authors.

The proposed co-design strategy effectively addresses expert routing and communication overheads in MoE-style LLMs, however, it is highly specialized to this sparse model family. The system assumes hard top-k expert gating and token-level dispatch, which do not generalize to dense autoregressive models (e.g., GPT-3, LLaMA), diffusion models, or state-space architectures (e.g., Mamba) that lack modular sparse execution patterns. Additionally, the hardware assumptions—modular chiplet-based designs with a Network-on-Package interconnect—limit applicability to a narrow class of accelerators. Real-world systems such as monolithic GPUs, wafer-scale engines (e.g., Cerebras), and mobile AI chips follow vastly different architectural and memory paradigms. The lack of runtime adaptability in expert scheduling and coarse treatment of memory hierarchies further constrain its deployability. A discussion on these generalization limitations is absent.

Regarding the empirical evaluation. There is a lack of baselines beyond static scheduling. The main baseline is a static MoE routing strategy or uniform tiling. It would be good to add comparisons against DeepSpeed-MoE or Tutel routing strategies and Existing expert placement heuristics (e.g., token-balanced sharding, load-aware dispatch). The existing evaluation is not enough to claim that Mozart significantly outperforms state-of-the-art MoE runtimes.

Limited System Scaling Study. The paper does not analyze how Mozart behaves under increasing number of chiplets, varying interconnect topologies (e.g., mesh vs. ring) and realistic memory capacity constraints. Such scaling or sensitivity analysis can showcase if the benefits hold at larger or smaller deployment scales.

While latency and utilization are discussed, the paper completely omits energy or area modeling. Given the focus on chiplet hardware, this is a notable omission—MoE communication patterns can be power-intensive.

The compute-communication overlap seems similar to existing works like DualPipe from DeepSeek and TileLink from ByteDance.

(d) Provide the most important reasons for your decision to accept/reject. For spotlights or orals explain why the paper stands out (other than by high scores or popularity trends).

As the title of the conference is Neural Information Processing *Systems*, it is good to see papers exploring the forefront of the computational substrate for these systems. One of the drivers of the current renaissance of machine learning has been the massive increase in computational capacity due to high-performance GPU systems. Wafer-scale systems seem to be the next frontier that will drive the next wave of development in machine learning. The paper presents, and justifies/validates, a new chiplet based wafer scale substrate supporting the next generation of machine learning systems.

(e) Summarize the discussion and changes during the rebuttal period. What were the points raised by the reviewers? How were each of these points addressed by the authors? How did you weigh in each point in your final decision?

The rebuttal discussion was extensive, and the authors did a lot of work in addressing the issues raised by the reviewers.

One key issue that was clarified in the rebuttal discussion was the contributions made on the hardware side. As doing FPGA, ASIC or full custom VLSI is premature (and expensive) at this stage, the detailed cycle-accurate RTL simulations are sufficient to establish the benefits and feasibility of the approach.

As requested by reviewer 6v38 the authors provided extensive comparisons on an advanced server with 8x NVIDIA A100 GPUs (each with 80GB HBM) to mimic Mozart, which features 4 chiplet clusters and has a similar total power consumption. This comparison was non-trivial and must have involved a great deal of additional work on the part of the authors.